# Land reclamation pattern and environmental regulation guidelines for port clusters in the Bohai Sea, China

Gaoru Zhu[1]☯, Zhenglei Xie  [2]☯*, Honglei Xu[1]*, Minxuan Liang[3], Jinxiang Cheng[1], Yujian Gao[1], Liguo Zhang[1]

1 Laboratory of Transport Pollution Control and Monitoring Technology, Transport Planning and Research Institute, Ministry of Transport of the People's Republic of China, Beijing, China, 2 College of Marine Science and Engineering, Nanjing Normal University, Nanjing, Jiangsu, China, 3 South China Institute of Environmental Sciences, Ministry of Ecology and Environment, Guangzhou, Guangdong, China

☯ These authors contributed equally to this work.
* 76010@njnu.edu.cn (ZX); yibin.zgr@163.com (HX)

## Abstract

Coastal land reclamation (CLR), particularly port reclamation, is a common approach to alleviating land shortages. However, the spatial extent, percentages, and processes of these newly reclaimed ports are largely unknown. The Bohai Sea is the most concentrated area of port reclamation worldwide. Thus, this study addresses the changes in the different coastline types and port reclamation process in the area. The reclamation area of the 13 ports in the Bohai Sea in 2002–2018 was 2,300 km$^2$, which decreased the area of the sea by 3%. The natural coastline length in Tianjin decreased by 47.5 km, whereas the artificial coastline length increased by 46.6 km. Based on the port boundary, however, only 26.3% of the reclaimed areas have been used for port construction, which concentrates in the Tianjin and Tangshan ports. The ratio of built-up area within the ports is only 32.5%, and approximately 48.3% of the reclaimed areas have no construction projects. The port land reclamation in the Bohai Sea has been undergoing periods of acceleration, peak, deceleration, and stagnation since 2002. Hence, future port reclamation should not be totally prohibited, and fine management should be conducted based on the optimization of the reclaimed port area. The innovation of this research is its analysis of the port internal land use pattern, the percentage of built-up area in the ports, and the sustainability of port reclamation policies. The findings have vital implications for scientifically regulating the spatial pattern and exploring the utility of port reclamation.

## Introduction

Coastal areas worldwide host 3 billion people and contribute to approximately 70% of the world's gross domestic product (GDP) due to easy access to efficient transportation and trade facilities [1, 2]. Ports in coastal areas have become fundamental nodes and choke points of global trade networks, which are believed to be engines for economic growth and exchanges,

**Funding:** This research was founded by the National Key Research and Development Plan

(Grant No. 2017-YFC-1405500), National Natural Science Foundation of China (No. 41601105, 41861041), China Scholarship Council (CSC No. 201708360067).

**Competing interests:** The authors have declared that no competing interests exist.

national prosperity, and social sustainable development [3, 4]. Ports across the world are the only transportation facility that drives the rapid economic development and prosperity of countries [5, 6]. China has been a main port country over the last few decades, with 30,388 berths by the end of 2016, of which 2,317 host ships of more than 10,000 tons [7]. Seven 100-million-ton container ports in mainland China are among the world's top ten ports in global throughput, through port reclamation intended for construction [3, 5, 8]. Port clusters are the general trend of international port development and fulfill increasingly distinct functions in the economic development of coastal areas [9]. The higher the port clustering development level, the greater the per capita GDP growth rate [8].

The shortage of land resources has been a bottleneck in high-quality economic development since the late 1990s [1]. With rapid regional economic and social development and increasing population in coastal areas, reclamation has been regarded as an effective and potential approach to resolve the land shortage for living and development [10]. The most prominent type is the reclamation of ports, especially since most of the ports worldwide are constructed by occupying coastal and marine space [5, 8, 11]. CLR is conducted through the construction of dikes or earthwork landfilling outside coastlines, thereby transforming natural areas to artificial terrestrial surfaces that extend toward seas [12–16]. Port development significantly depends on coastal land reclamation (CLR) for expanding the requisite space. Additionally, China not only has the largest port population but also has the largest number of port reclamation areas worldwide [5]. Over 70% of the coastal reclamation occurred in the northern coastal region of China [14]. Large-scale CLR, which has basically ceased in developed countries but remains popular in developing countries, has fundamentally mitigated the tense situation of urban and industrial land use [17–20]. The total amount of reclamation projects and the scale of individual reclamation projects in China rank first in the world [7]. The evolution stages of CLR have transformed from the traditional salt production saltpans in the 1950s and mid-1960s, agricultural purposes in the 1960s and 1970s, and mariculture in the 1980s and 1990s to ports, industrial park, adjacent harbor industries, and urban construction in the 21$^{st}$ century [5, 7, 19, 25]. CLR has extended coastlines in mainland China from 15,612.6 km in 1980 to 18,604.4 km in 2015, with the coastline type changing from natural to artificial one [7].

CLR leads to substantial benefits, but it is widely criticized for causing irreversible changes in the natural properties of coastal regions [21–25]. Port CLR can lead to the loss of natural habitats, changes in ecosystem services, and substantial reductions in mudflats and tidal influxes [25–27]. Previous studies have demonstrated that the per capita and urbanization level are closely associated with CLR, and the enormous economic and social benefits resulting from CLR may be the direct driving force for it [1, 14]. Current studies on port CLR focus on the effects of reclamation on the landscape patterns of coastal wetlands and sediment dynamics [28, 29] and the indicator systems of coastal reclamation intensity [30, 31]. The researchers applied high-resolution Gaofen-1 (GF-1) remote sensing images to reveal the component types and utilization traits of reclaimed areas in Yingkou, Liaoning Province, and established remote sensing monitoring technologies for the existing reclaimed land areas [32]. The researchers examined the landscape dynamics, development intensity, and driving forces in the reclaimed area in the Fangchenggang Port, Guangxi Zhuang Autonomous Region [33]. However, the application of remote sensing images to study port reclamation from the perspective of port internal land use structures remains insufficient.

The Bohai Sea Rim Area is an important economic center in northern China; the area has several national development strategies and is undergoing rapid urbanization and industrialization. Due to the rapid development of port clusters, the scale of land reclamation has been dramatically expanding. In 1980–2017, the reclaimed part of the Bohai Sea area, which is

located in the Bohai Sea mudflat coastal area, grew by 1,988.5 km$^2$ [34]. However, there are no ongoing construction projects in approximately 50% of the reclaimed area [34]. The reclaimed areas in the Caofeidian, Tianjin, and Dalian ports are the largest among all the ports in China [35]. The reclamation was launched in 2004 and to be completed in 2020, with 310 km$^2$ of land reclaimed from a shallow bank for deep water ports and for steel, chemical, electric works and nuclear power industries [35]. In 2018, the Chinese government suspended the acceptance and approval of reclamation projects and regional land reclamation plans. The main scientific questions are whether port CLR should restart to meet the rapid development of coastal economies and whether the negative effects of large-scale reclamation should be attributed to port CLR. Only few comprehensive studies have been conducted on the process, land use structures, and spatial distribution of CLR at the regional scale. Additionally, the reclamation trends of planned port clusters and environmental regulation schemes remain unclear. This research aims to apply Landsat imagery data acquired in 2002 and 2018, documents from the China Port Statistical Yearbook, and planning data to measure the changes in the coastline length and the coastal reclamation area and assess the spatial extent and process of CLR in the Bohai Sea, China.

## Study area and research method

### Ethics statement

Our study area is located in the Bohai Sea, which is owned by the Chinese government. This study did not involve endangered or protected species and no specific permissions were required for the locations/activities in this study. The specific locations in the present study are shown in Fig 1.

### Study area

The Bohai Sea, with an area of 77,000 km$^2$, is a typical semi-enclosed coastal sea and with a mean depth of 12.5 m (Fig 1). The coasts of Bohai Bay, the Yellow River Delta, and the northern portion of the Liaodong Bay are silty and muddy with a low slope ranging 0.1%~0.6%. The geomorphic units consist of alluvial plains, coastal plains, mudflats, and submarine slopes. The coast of the northern part of the Shandong peninsula and the western part of the Liaodong peninsula are rocky coasts. The soil types are coastal saline-alkali soil with high salinity and pH, brown earth, and marshy and rice soil. The abundant sediments are beneficial for land reclamation. The area is located in semi-humid continental monsoon climate area and the warm temperate zone. The average annual temperature is 11 ~ 13°C and the mean annual precipitation is 600 ~ 900 mm, which most of the rain occurring in summer. The natural vegetation includes herbaceous plants and shrubs with low forest cover, and salt meadows dominate the biological community. Moreover, there are several national natural reserves, including the Tianjin Palaeocoast and Wetland Natural Reserve, Laotieshan Natural Reserve, Dalian *Phoca Largha* National Natural Reserve, Bohai Shell Island and Wetland National Natural Reserve. The main habitats are tidal mudflats, saltpans, and shrimp ponds, and a sea wall separates the saltpans and shrimp ponds from the mudflats. The area includes many rivers and shallow tidal flats, which form a special coastal wetland landscape. Meanwhile, the coastal area of the Bohai Sea is an important habitat for endangered birds. There are salt works, wharfs, oil fields, and industrial areas along the coast.

The port clusters include Yingkou, Panjin, Jinzhou, Huludao, Qinhuangdao, Tangshan, Tianjin, Huanghua, Binzhou, Dongying, Weifang, Dalian, and Yantai. Since the 1990s, ports have been rapidly developing, forming a port cluster with more than 10 ports [36, 37]. The harbor area has the largest proportion in China, highlighting the role of the harbor and shipping

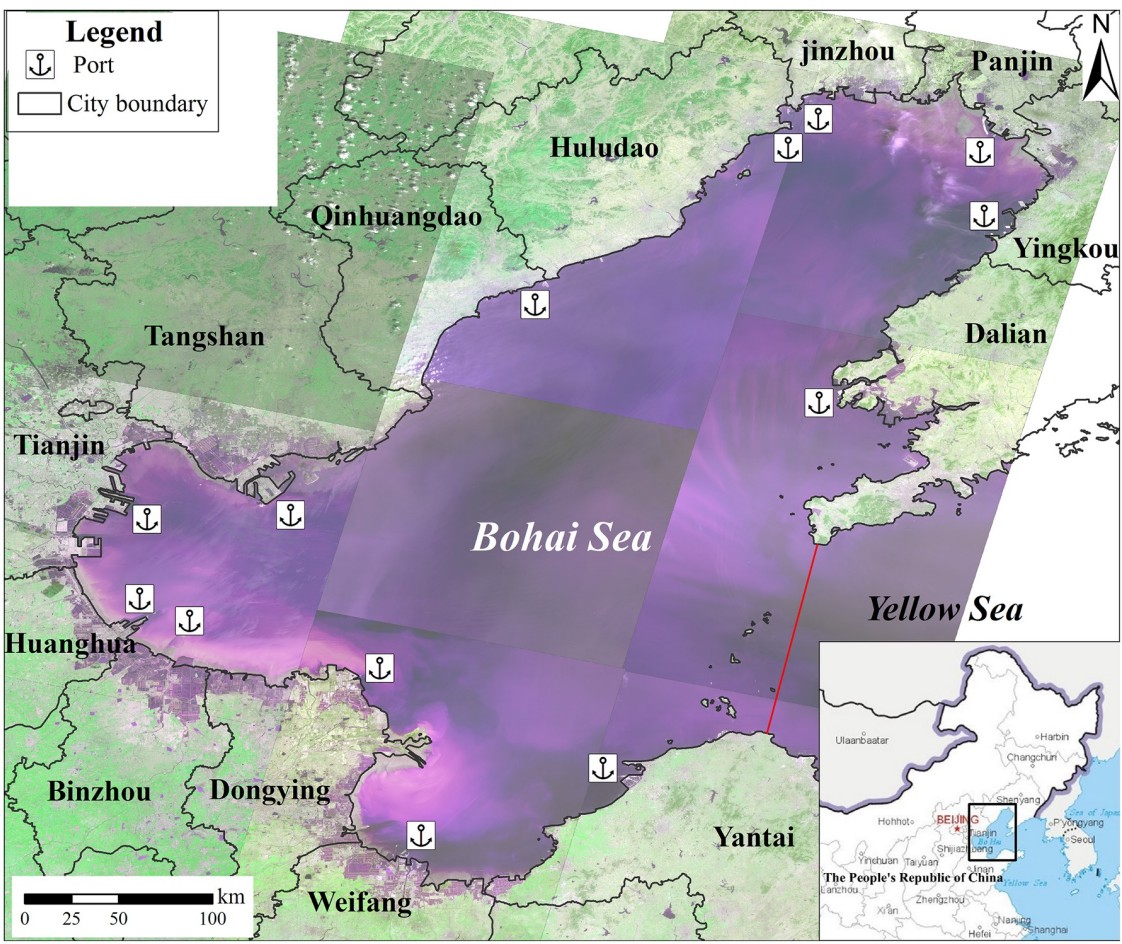

**Fig 1. Location map of the study area.** It includes the Liaodong Bay, Bohai Bay and Laizhou Bay. The region is located in Liaoning, Tianjin, and Hebei and Shandong. The coastal area of the Bohai Sea has many ports and has been experiencing rapid industrialization and urbanization.

centre in North China as well as the adjacent harbor industry agglomeration area [21]. Approximately 307 km² of coastal wetland has been reclaimed in the Tianjin Binhai New Area over the past 10 years, which Tianjin port locates this area. These ports namely Tianjin port, Tangshan port located in Caofeidian New Area have the largest artificial reclamation area planned in the 11[th]-Five-Plan of China. In 2018, the cargo throughput of the port clusters was 2.5 billion tons, which was five times than that in 2002. Port cargo throughput in Tangshan, Tianjin, and Yingkou ports was relatively higher, while the growth rate of cargo throughput in Binzhou, Panjin, and Dongying ports was relatively faster than other ports. Through port reclamation, the Tianjin Binhai New Area has turned a desolate coast into a modern metropolitan area. Tianjin port focus on the development of port and coastal industries, with a total planned reclamation area of 2,270 km². The Caofeidian New Area in Tangshan port has been one of the national comprehensive development strategies in 2010 and is promoting the construction of the Caofeidian International Eco-city. The reclamation intensity from 2001 to 2012 demonstrated a continuous increment [38].

## Research method

Cloud-free Landsat 5 TM and Landsat 7 ETM+ remote sensing images with a spatial resolution of 30 m were downloaded from the Geospatial Data Cloud (http://www.gscloud.cn) and the United States Geological Survey website (http://glovis.usgs.gov). Seventeen images acquired in 2002 and 2018 were used to identify the coastline changes and land use classifications (Table 1). The Univeral Transverse Mercator (UTM) was the projection reference system, and the reference ellipsoid was WGS 84. Preprocessing steps, such as band combination, geometric correction, radiation correction, atmospheric correction, clipping and mosaic creation, were performed using the software ERDAS Imagine 12 [21]. A visual interpretation method was applied to identify the land use/land cover change process. The reference maps were the topographic map of 1981, the land use database compiled by the Institute of Geographic Science and Natural Resources (Chinese Academy of Sciences), and the land use plan status map. The Transport Planning and Research Institute (Ministry of Transport)'s master plan for port clusters in 2030 was also obtained and combined with the preceding data to constrain the boundary of reclamation and changes in the exploration intensity of port clusters. The interpreted images were verified by field investigation in July 2019 based on identified points of land use with a portal GPS.

Based on the Landsat images of 2002 and 2018, the location of the land-sea demarcation line was extracted to analyze the changes in the coastline and the distribution of the reclaimed areas in the Bohai Sea in 2002–2018. The coastline types were natural, semi-natural, and artificial. A semi-natural coastline is a reclamation coastline with certain ecological functions, such as salt fields and aquaculture ponds. The port boundary was used to monitor the reclamation of port clusters and land use status within the ports. The changes in national policies were summarized to evaluate the causes of the changes in port clusters and to suggest environmental regulation strategies. Then, the China Port Statistical Yearbook of 2002–2018 was used to identify the relationship between cargo throughput and CLR. In July 2019, a field survey of the reclamation status in the Bohai Sea was conducted; the managers of the port departments were

**Table 1. Path/Row and date of Landsat remote sensing images used in the study.**

| Year | Path/row | Date |
|------|----------|------|
| **2002** | 120/032 | 2002-07-12 |
| | 120/033 | 2002-05-25 |
| | 120/034 | 2002-10-16 |
| | 121/032 | 2002-08-20 |
| | 121/033 | 2002-09-21 |
| | 121/034 | 2001-07-16 |
| | 122/032 | 2002-05-23 |
| | 122/033 | 2002-11-15 |
| | 122/034 | 2002-08-11 |
| **2018** | 120/032 | 2018-04-19 |
| | 120/033 | 2018-04-19 |
| | 120/034 | 2018-04-19 |
| | 121/032 | 2018-04-19 |
| | 121/033 | 2018-04-19 |
| | 121/034 | 2018-04-19 |
| | 122/032 | 2018-04-19 |
| | 122/033 | 2018-04-19 |

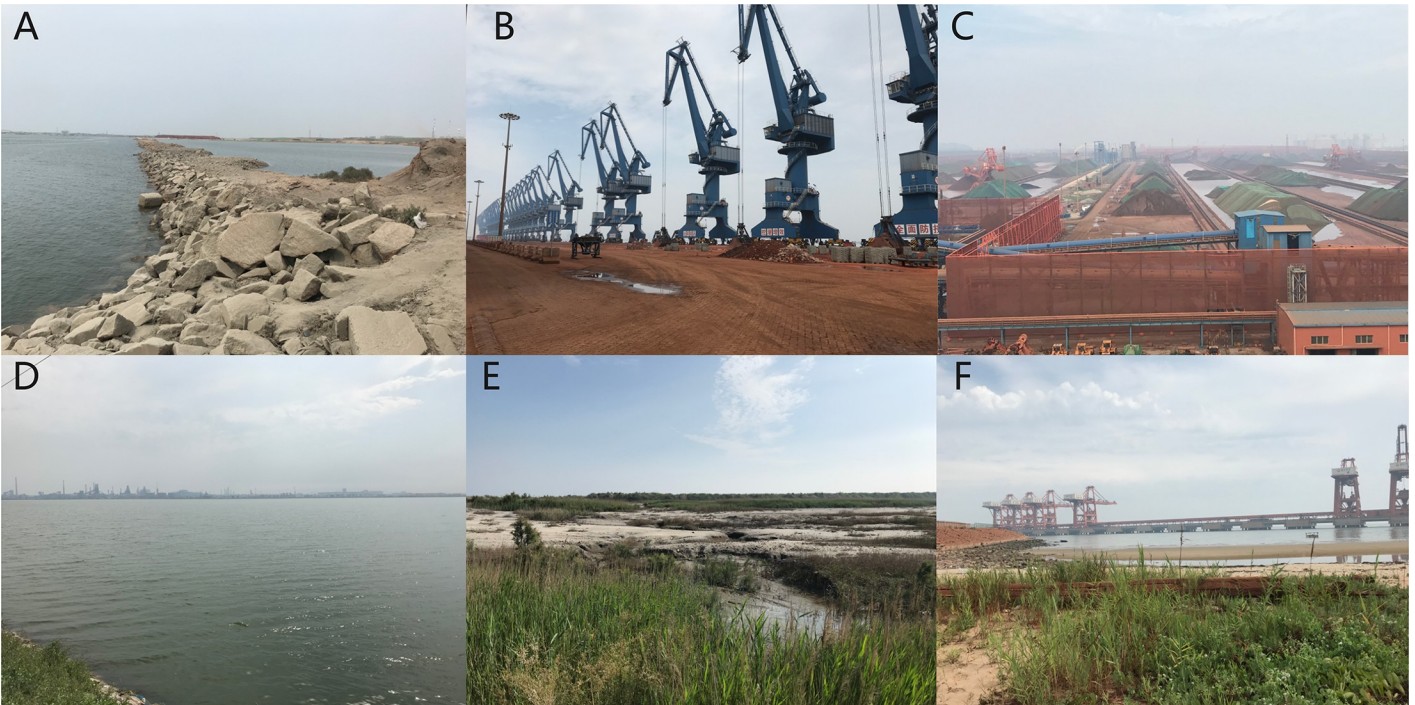

**Fig 2. Typical landscape in the reclaimed area in the Bohai Sea.** In 2009, a coastal survey was conducted and identified different reclamation process. (A) In the beginning of reclamation; (B) Port land use; (C) Storage yard; (D) Newly enclosing sea; (E) Unused built-up; (F) Vegetation within ports.

interviewed about the policy evolution, land use changes and existing issues in the reclaimed area (Fig 2).

The port reclamation area and the length of the different coastline types were used in Tabulate Area from the toolbox of ArcGIS 10.2. The land use composition in the ports were classified as built-up, forest and grassland, enclosing sea, unused built-up, saltpans and shrimp fields. The proportion of built-up areas to the total port area was the sum of the percentages of the built-up, forest and grassland, enclosing sea, unused built-up, saltpans and shrimp fields. The reclamation intensity index, which referred to the reclaimed area within a coastline length, was used to quantitatively assess the scale and intensity [39].

$$\text{Reclamation intensity index (ha/km)} = \frac{Reclamation\_area}{Coastline\_length}. \tag{1}$$

Reclamation area (ha) is the reclaimed area and Coastline length is the coastline length (km).

$$\text{Utilization efficiency of port (t/km}^2) \frac{Port\_throughput}{Port\_area}. \tag{2}$$

Port_throughput is the throughput of each port, and Port_area is the total port area [40]. With the China Ports Statistical Yearbook, the changes in port land use efficiency could be identified from the port throughput from 2002 to 2018 and the port land area. As for the changes in the reclamation policies, related articles were used to identify the policy trends of reclamation.

**Table 2. Changes of different coastlines length and reclamation areas in the Bohai Sea during 2002–2018.**

| City name | Coastline length (2002) (km) | | | Changes of coastline length (2002–2018) (km) | | | Total reclamation area 2002–2018 (km²) |
|---|---|---|---|---|---|---|---|
| | Natural | Semi-natural | Artificial | Natural | Semi-natural | Artificial | |
| Binzhou | 0 | 88.6 | 0 | 0 | 0 | 0 | 83.6 |
| Dalian | 281.1 | 233.0 | 16.4 | -148.5 | +50.7 | +97.8 | 276.2 |
| Dongying | 112.8 | 236.0 | 57.8 | -12.9 | -41.5 | +54.4 | 329.7 |
| Huanghua | 0 | 54.0 | 17.5 | 0 | -13.5 | +13.5 | 93.1 |
| Huludao | 67.3 | 117.1 | 51.5 | -49.6 | +6.0 | +43.6 | 59.8 |
| Jinzhou | 7.0 | 188.3 | 0 | -7.0 | +1.8 | +5.2 | 152.1 |
| Panjin | 0 | 30.5 | 0 | 0 | -14.7 | +14.7 | 119.4 |
| Qinhuangdao | 34.0 | 63.8 | 34.4 | -4.1 | -2.2 | +6.4 | 23.6 |
| Tangshan | 2.2 | 200.5 | 7.3 | 0 | -44.6 | +44.6 | 334.4 |
| Tianjin | 47.5 | 46.9 | 55.4 | -47.5 | +0.9 | +46.6 | 340.4 |
| Weifang | 14.3 | 151.0 | 0 | -10.1 | -40.1 | +50.2 | 271.3 |
| Yantai | 6.9 | 120.5 | 89.9 | -6.9 | -12.9 | +19.8 | 132.8 |
| Yingkou | 25.5 | 65.2 | 11.2 | -18.9 | -29.4 | +48.3 | 83.7 |
| Sum | 598.5 | 1595.3 | 341.5 | -305.5 | -139.5 | +444.9 | 2,300.0 |

Note: Semi-natural coastline refers to the reclamation coastline with certain ecological functions, such as salt fields and aquaculture ponds.

## Results

### Changes of reclamation areas and coastline lengths of port clusters in the Bohai Sea

The reclaimed area in 2002–2018 was 2,300 km², and it decreased the area of the Bohai Sea from 77,500 to 75,200 km². The reclaimed areas in ports such as Tianjin, Tangshan, and Dongying were more than 300 km². The natural coastline length in Tianjin decreased from 47.5 to 0 km, whereas the artificial coastline length increased from 55.4 to 102.0 km (Table 2). Additionally, the natural coastline length in Dalian decreased by 148.5 km, and the artificial coastline length increased by 97.8 km (Fig 3).

### Port reclamation areas and port land use efficiency in the Bohai Sea

The port area in the Bohai Sea increased from 335.88 km² in 2002 to 950.27 km² in 2018; the growth of the Tianjin, Tangshan, and Huanghua ports was larger than that of the other ports. The area of Tianjin Port increased from 41.1 to 201.03 km², that is, by 3.9 times, and the area of Tangshan Port expanded from 63.26 to 177.99 km², i.e., by 1.8 times (Table 3). The reclaimed areas in Huludao, Dongying, and Weifang ports were relatively smaller. However, only 26.3% of the reclamation area was used for port construction based on port boundaries. The other areas were used as adjacent port industries, for urban construction, and for saline and aquaculture development. The percentages of the reclamation areas in the Huanghua, Tianjin, and Tangshan ports were 96.6%, 47.0%, and 34.3%, respectively. The percentages of the Dongying, Weifang, and Huludao ports were less than 10%. According to the China Port Statistical Yearbook of 2002–2018, the port clusters' cargo throughput increased sharply from 377.38 to 2,550 million tons and the total port area increased from 335.88 to 950.27 km². Furthermore, the port land use efficiency of the Bohai Sea in 2002–2018 increased from 1.12 to 2.68 million ton/km² (a growth rate of 139%), and the port land in the reclaimed area was fully utilized. Most of the port land use efficiency increased evidently; the current land use efficiency

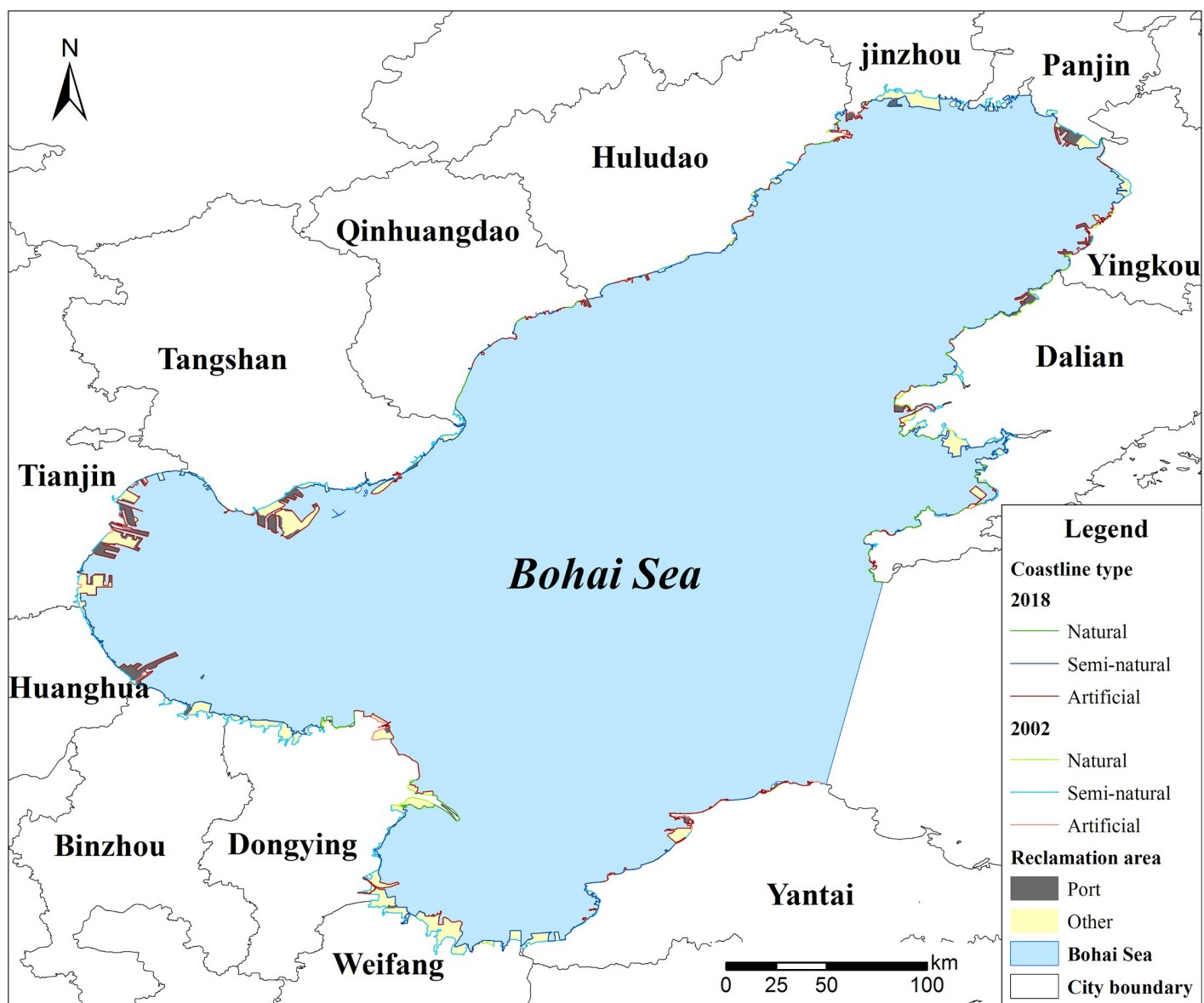

**Fig 3. Changes of coastline length of natural, semi-natural, artificial types, reclamation area in the reclaimed area in 2002–2018 in the Bohai Sea.** From the perspective of spatial distribution, the reclaimed areas in the main ports were more than 300 km². The natural coastline length in Tianjin decreased sharply and artificial coastline length increased evidently.

and efficiency increment ratio in the Dongying and Yingkou ports are relatively high, whereas those in the Tianjin and Jinzhou ports are relatively low. The present land use efficiency in the Panjin and Binzhou ports is low, but the efficiency increment rate is considerable.

## Areas of different land use types and percentages of build-up of port clusters in the Bohai Sea

The land use types of the port clusters are built-up, forest and grassland, newly enclosing sea, unused built-up, saltpans, and shrimp fields. In 2018, the built-up area within the port boundary was 309 km² (32.5%), as tabulated in Table 4. The percentages of the built-up area in all

**Table 3. Changes of port reclamation area, port built-up percentage, throughput, and land use efficiency during 2002–2018 in the Bohai Sea.**

| Port name | Port reclamation Area (km²) | | Built-up percentage (%) | Throughput (10⁴ ton) | | Efficiency (10⁴ t/km²) | |
|---|---|---|---|---|---|---|---|
| | 2002 | 2018 | | 2002 | 2018 | 2002 | 2018 |
| Binzhou | 11.44 | 24.28 | 15.4 | 26.8 | 3140 | 2.3 | 129.3 |
| Dalian | 71.81 | 141.29 | 25.2 | 2570 | 10813 | 35.8 | 76.5 |
| Dongying | 3.15 | 9.86 | 2.0 | 25 | 5,825 | 7.9 | 590.8 |
| Huanghua | 31.41 | 129.96 | 96.6 | 1,801 | 28,771 | 57.3 | 221.4 |
| Huludao | 0.93 | 4.25 | 5.6 | 153 | 1,640 | 164.5 | 385.9 |
| Jinzhou | 8.67 | 36.27 | 18.1 | 1,404 | 10,960 | 161.9 | 302.2 |
| Panjin | 21.17 | 76.92 | 46.7 | 40 | 4,091 | 1.9 | 53.2 |
| Qinhuangdao | 20.53 | 33.35 | 54.2 | 11,167 | 23,119 | 543.9 | 693.2 |
| Tangshan | 63.26 | 177.99 | 34.3 | 1,465 | 63,710 | 23.2 | 357.9 |
| Tianjin | 41.1 | 201.03 | 47.0 | 12,900 | 50,774 | 313.9 | 252.6 |
| Weifang | 0 | 8.42 | 3.1 | 244 | 4,656 | | 553.0 |
| Yantai | 24.7 | 40.64 | 12.0 | 2,936 | 10,500 | 118.9 | 258.4 |
| Yingkou | 37.71 | 66.01 | 33.8 | 3,006 | 37,001 | 79.7 | 560.5 |
| Sum | 335.88 | 950.27 | 26.3 | 37,737.8 | 255,000 | 112.4 | 268.3 |

Note: Percentage: reclamation area from sea used for ports in all the reclaimed area (%).

ports showed noticeable differences. For example, the percentages of the built-up area in the Yingkou and Yantai ports were 59.7% and 57.5%, respectively, whereas the percentages in the Binzhou and Weifang ports were both less than 10% (Fig 4). The unused built-up area within the ports was 458.6 km², and the area of the newly enclosing sea was 124.4 km², which accounted for 61.4% of the entire reclaimed area. The unused built-up areas in the Tangshan, Tianjin, and Huanghua ports were 115.5, 78.2, and 75.2 km², respectively, indicating substantial room for development. By contrast, the unused built-up area in the Huludao, Dongying, and Weifang ports was relatively smaller and can only accommodate limited future development. The newly enclosing sea in the Dalian and Binzhou ports was larger than in the other ports, according to current reclamation policies.

**Table 4. Areas of different land use types within the port boundary in the Bohai Sea in 2018 (Unit: Km²).**

| Port name | Built-up | Forest and grassland | Newly enclosing sea | Unused built-up | Saltpans and shrimp fields | Percentage of built-up (%) |
|---|---|---|---|---|---|---|
| Binzhou | 1.8 | 0.0 | 22.5 | 0.0 | 0 | 7.5 |
| Dalian | 14.5 | 5.8 | 29.3 | 63.7 | 28 | 10.3 |
| Dongying | 2.8 | 0.0 | 2.5 | 4.6 | 0 | 28.4 |
| Huanghua | 34.9 | 0.0 | 19.9 | 75.2 | 0 | 26.8 |
| Huludao | 2.3 | 0.0 | 0.5 | 1.5 | 0 | 53.4 |
| Jinzhou | 13.3 | 0.0 | 9.9 | 13.1 | 0 | 36.5 |
| Panjin | 23.3 | 0.0 | 6.8 | 46.8 | 0 | 30.2 |
| Qinhuangdao | 15.0 | 0.5 | 3.1 | 14.7 | 0 | 45.1 |
| Tangshan | 35.4 | 0.0 | 3.6 | 115.5 | 23.5 | 19.9 |
| Tianjin | 102.2 | 0.2 | 20.4 | 78.2 | 0 | 50.8 |
| Weifang | 0.8 | 0.0 | 0.3 | 7.3 | 0.1 | 9.9 |
| Yantai | 23.4 | 0.1 | 4.2 | 12.8 | 0 | 57.7% |
| Yingkou | 39.4 | 0.0 | 1.4 | 25.2 | 0 | 59.7% |
| Sum | 309.0 | 6.6 | 124.4 | 458.6 | 51.6 | 32.5% |

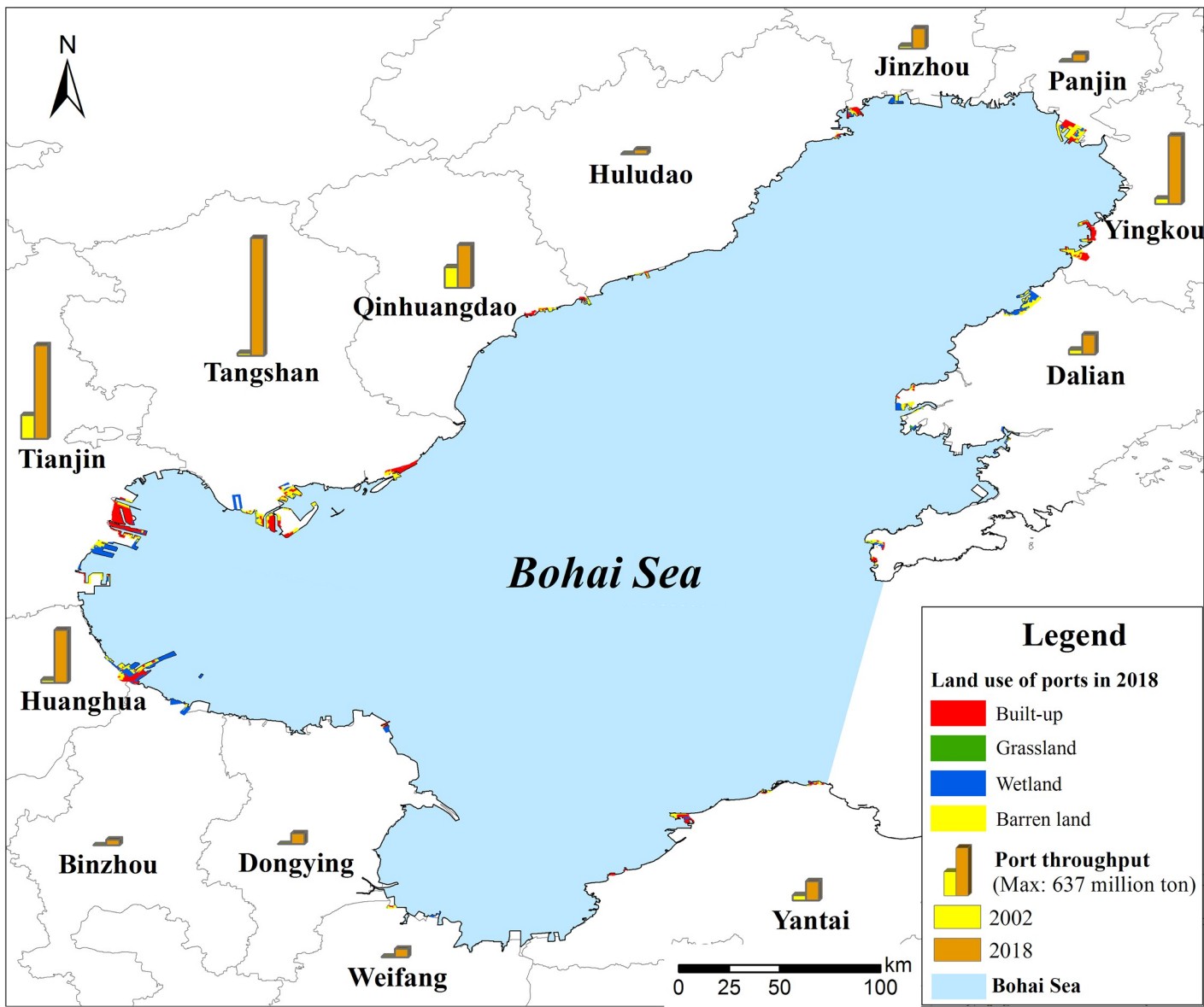

**Fig 4. Land use types within the reclaimed area and throughput changes of port clusters in the Bohai Sea.** The land use types were classified into built-up, forest and grassland, newly enclosing sea, and other land use types. The proportion of built-up in all ports showed obvious differences.

### Area and intensity of reclamation plan of port clusters in 2030

According to the port development plan in the Bohai Sea, the port area will continue to expand, and will reclaim an area of approximately 620.4 km$^2$ by 2030 (Fig 5). The planned reclamation areas in the Binzhou, Tangshan, and Huanghua ports are 122.8, 91.1, and 79.8 km$^2$, respectively, which account for a relatively large region (Table 5). By contrast, the planned reclamation areas in the Panjin and Dongying ports are smaller than 10 km$^2$. Based on the coastline length of each port, the planned annual reclamation intensity in 2018–2030 should be 2.04 ha km$^{-1}$, which reflects an increment of 37% from the size in 2002–2018. The reclamation intensity in the Binzhou and Huanghua ports is relatively high, greater than 8 ha km$^{-1}$, but the

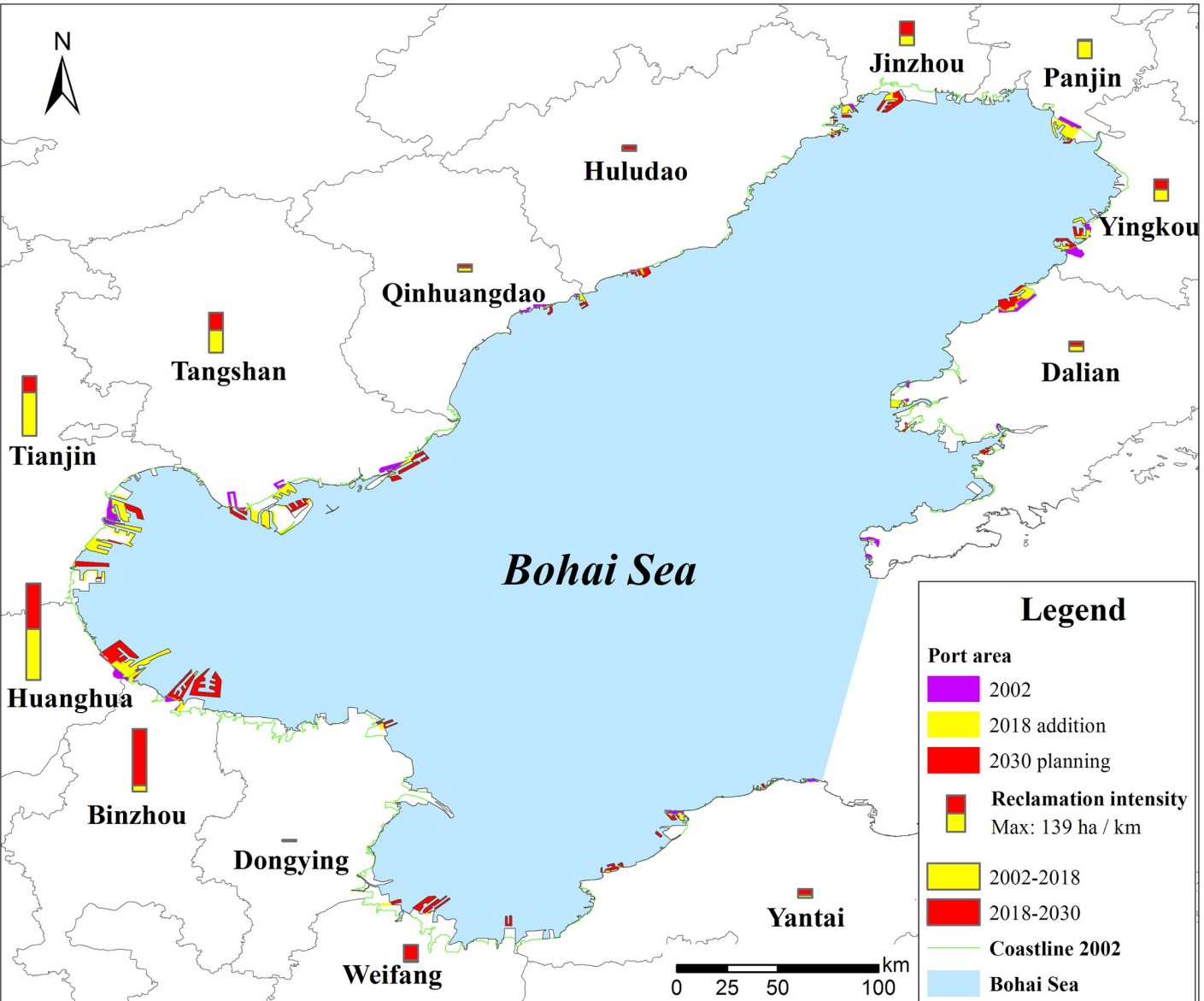

**Fig 5. Future trends of reclamation intensity of port clusters in the Bohai Sea.** The port reclamation will continue to expand. However, different ports have different reclamation intensity. For example, the planned reclamation area of Binzhou, Tangshan, and Huanghua ports is relatively larger, however, the number in Panjin and Dongying port is less than 10 km².

increment rate of the planned annual reclamation in the Binzhou, Huludao, and Weifang ports is relatively larger.

## Discussion

### Port reclamation areas and coastline length

Due to the construction of a large number of coastal facilities, the total length of coastlines in mainland China has increased from 15,612.6 km in 1980 to 18,604.4 km in 2015, which then led to the constant shifting of coastline types, from natural to artificial type [41]. Meanwhile, natural coastlines have been continuously reduced from 1980 to 2015 [8]. The coastlines in

**Table 5. Planned area and intensity of port clusters reclamation in the Bohai Sea in 2018–2030.**

| Port name | Planned reclamation area in 2018–2030 (km$^2$) | Annual reclamation intensity per unit coastline (ha/km) | | |
|---|---|---|---|---|
| | | 2002–2018 | 2018–2030 | Variance (%) |
| Binzhou | 122.8 | 0.91 | 11.57 | 1175% |
| Dalian | 60.3 | 0.81 | 0.94 | 16% |
| Dongying | 8.2 | 0.10 | 0.17 | 62% |
| Huanghua | 79.8 | 7.87 | 9.31 | 18% |
| Huludao | 26.4 | 0.10 | 1.01 | 960% |
| Jinzhou | 38.7 | 1.52 | 2.84 | 87% |
| Panjin | 4.0 | 2.68 | 0.25 | -91% |
| Qinhuangdao | 10.6 | 0.61 | 0.67 | 10% |
| Tangshan | 91.1 | 3.42 | 3.62 | 6% |
| Tianjin | 59.9 | 6.68 | 3.34 | -50% |
| Weifang | 61.6 | 0.32 | 3.11 | 875% |
| Yantai | 31.8 | 0.46 | 1.22 | 166% |
| Yingkou | 25.2 | 1.78 | 2.11 | 19% |
| Sum | 620.4 | 1.49 | 2.04 | 37% |

Tianjin's urban and industrial land areas, especially Tianjin port construction and adjacent industry development, extended seawards from 1996 to 2015 in the form of an irregular polygon, which increased the coastline length, especially after Tianjin Binhai New Area was included in the national overall development strategy [7, 8, 13]. In this study, the coastal length and different type coastline lengths were identified. The coastline types are divided into natural, semi-natural, and artificial ones and the percentage of artificial one among all the types increased from 36.98% in 2002 to 68.09% in Tianjin Port in 2018. Therefore, the results are almost same with Xu's research in 2016. Xu found that the percentage of the artificial coastline length has increased from 32.3% in 1980 to 68.5% in 2015 [41]. As for semi-natural of saltpans, the total area of Changlu saltpans in Tanggu, Hangu salt pans in Tianjin in 1949–1965 expanded from 216.0 to 6,580.0 km$^2$, thereby increasing the coastline length [25, 42]. The change in natural coastline is permanent and irreversible, and the transformation of coastlines from natural to artificial one has led to the changes of water flow direction, landform, and ecosystem structure of the coastal zone, the loss of coastal wetlands and the destruction of coastal ecosystem biodiversity [8, 14, 43]. Thus, even if a certain type of coastline is suitable for port location, coastline resource conservation should be guaranteed during the process of port reclamation construction.

Based on both historical statistics and satellite images, it indicated that the total area of reclaimed land increased from 8,241 km$^2$ to 13,380 km$^2$ between 1990 and 2008, an average increase of 285 km$^2$ annually [14, 36]. More than 90% of the reclaimed land has been developed for commercial or residential use. Tianjin Port focus on the development of port and coastal industries with a total planned reclamation area of 2,270 km$^2$ [44]. In our study, in 2002–2018, the total reclamation area is 201.03 km$^2$ in Tianjin port. The overall area of reclamation in 2002–2018 was evidently large, and port clusters reclaimed 605.8 km$^2$, which accounts for almost 1% of the total area of the Bohai Sea. The planned reclamation area in China, however, is larger than the actual demand, and incorporating the management idea of total control and intensive utilization is urgent [44]. Carrying out ecological construction and management of port reclamation from the control of reclamation scale is an important path to achieve the sustainable development in coastal area. China has attached great importance to

reclamation management and established a management system focusing on total amount control and intensive management [45].

## Percentage of build-up area, different land use, and reclamation intensity

The land use types within the ports consists of built-up, forest and grassland, newly enclosing sea, unused built-up, and saltpans and shrimp field. As for the percentage of built-up within the ports, the largest land use types are built-up in 2018 while the land use type with largest area in Tangshan port is unused built-up in 2018. Although ports have expanded development possibilities through large-scale reclamation, this has led to problems related to newly enclosing sea and unused built-up area. In China, the proportion of port reclamation areas to the total reclamation area since the 1980s is 8.1%, which is far smaller than the proportion of saltpans, aquaculture areas, and industrial land. For the densest port reclamation area of the Bohai Sea, the percentage of the reclaimed area used for port construction in 2002–2018 was only 25%. Our findings demonstrated that the built-up area within the port boundary was 309 km$^2$; this figure was equivalent to only 32.5% in 2018, where 458.6 and 124.4 km$^2$ of the reclaimed sea were unused built-up areas and enclosing sea, respectively. Moreover, the reclaimed area in the Dalian and Tangshan ports have saltpans and shrimp farms, which can provide room for further port development [13]. In China, a total area of 1,400 km$^2$ has been reclaimed but is not yet being used; this is a planned adjacent industrial area and a new industrial city. A large part of reclaimed land within the ports remains under construction. The temporarily enclosing sea and unused built-up areas are being used for green land and enable ecological construction in the port areas; the idle land is not noticeable [8]. Many local governments tend to substitute port construction for other purposes, i.e., such as development of adjacent industries, urban dwellings, and tourism, but the reclaimed area within the port plan boundary is highly limited.

The Bohai Sea had the most intensive port reclamation in 2002–2018 in the world. The reclamation intensity of Tianjin in 2005–2010 is 20.2 ha/km and the reclamation intensity in Hebei is 17.4 ha km$^{-1}$, which the reclamation intensity was relatively larger [14]. In our study, the total reclamation intensity in Tianjin port in 2002–2018 is 6.68 ha km$^{-1}$, which is lower than the Tian's results in 2016. As for the total port cluster, the annual average reclamation intensity in 2002–2018 is 1.49 ha km$^{-1}$. Along with the area expansion of port reclamation, land use efficiency has greatly improved, which indicates that the saving and intensive utilization of ports have achieved positive results (Table 3). Due to the port construction period, the actual land use efficiency is relatively higher. Moreover, the current land use efficiency and the improvement status between the ports have relatively larger differences, showing that future port reclamation needs discretion and improvement of land use efficiency.

## Reclamation regulation strategies and economic development of port clusters

On a global scale, it has been estimated that about half of the population of the industrialized world lives within 1 km of the coast [46, 47]. Spurred on by a large population and fast-growing economies, coastal areas have aggressively expanded seafaring facilities and coastal industries and promoted urbanization, resulting in great demand for land area [25]. Because of this very strict arable land management policy, approval for development on arable land is essentially not possible [25, 48]. Reclamation is deemed as an effective method of addressing land scarcity and constructing ports. Extensive, high-intensity coastal reclamation activities deliver social and economic benefits; the contribution rate toward the GDP in coastal zones will reach 3%–7% [13, 14, 49, 50]. Ports are the only utilization type that depends on reclamation. Since

the 1980s, along with the fast development of economic globalization, port construction has been accelerating. In China, rapid adjacent port industry development and urbanization have significantly expanded the area of coastal reclamation, especially after 2000 [2, 14]. Moreover, advanced sand blowing technology makes it possible to reclaim 1 km$^2$ of land from the sea in about 20 days [25].

Port reclamation provides valuable development space for port expansion and establishes a basis for China to become an international water transportation power. Rapid and large-scale reclamation activities presently carried out in China's coastal areas have far exceeded the carrying capacity of the natural environment [25, 38, 54]. The seaward expansion of artificial coastlines weakens sediment dynamics and hydrodynamics, decreases the biodiversity and intertidal waterbird habitats, and reduce ecosystem services, leading to a reduction in the overall benefit of coastal reclamation [10, 25, 50–54]. The ecological service values of coastal wetlands in China will be CNY 20 million/km$^2$ each year, and port reclamation has directly decreased the ecological service value of occupied coastal wetlands [55]. The realized and potential environmental damages resulting from CLR are substantial. Some studies have only qualitatively analysed eco-system damage and geological disasters related to reclamation activities [35]. No have the environmental impacts of these large-scale reclamation activities aroused sufficient alarm. These impacts, however, must be fully assessed by comprehensive environmental studies and then minimized by the implementation of mitigation measures. Accordingly, the scale of ecological construction of seaports in China remains relatively small and needs to be promoted and innovated [8]. The environmental carrying capacity of the seaport clusters of Bohai Rim is low, however, the proportion of coastline utilization and the load of coastal exploitation is high [8]. Ports with high environmental carrying capacity mainly use natural bays in the process of engineering construction, with the relatively small area of artificial reclamation. Site selection for port ecological reclamation must be scientifically demonstrated and planned in advance, and the suitability of port reclamation must be explored from the perspective of engineering geology, hydrological conditions, and ecological environment [8]. A certain amount of distance is required for reclamation in coastal areas in certain regions of China to conserve natural coastlines and public water channels [44].

It is unacceptable to stop all coastal reclamation activities for environmental protection purposes. Guidelines should be created to provide practical environmental advice to developers who plan to undertake reclamation work in coastal areas. The ecological technology models and paths of ecological reclamation in other countries or regions may be adopted [8]. Moreover, land-ocean integration coordinates the spatial patterns of land and ocean, requiring either permission or prohibition of large-scale reclamation to reach a balance between industrial development and ecological conservation. Land-ocean integration requires unified arrangements of national territory spaces in coastal zones and reasonable development space plans for different industries. It reasonably demands an arrangement of the total amounts, timing sequences, and spatial distributions of land-ocean resource exploration and utilization. In order to adapt the new norms of economic development, optimizing land supply structure and ensuring the rational and healthy development of urban built-up area are very important [49]. Port construction should shift from reclaiming land from the sea to accomplishing land-ocean integration and from pursuing area expansion to fulfilling quality promotion, which focuses on port hinterlands for collecting and distributing traffic systems. The government should promote water-water transfers, coordinated river-and-sea transport, and coordinated railway-and-water transport.

## Policy evolution and stages of CLR in China

The reclamation policy in China since the 1970s has undergone the transition through "free support before 1980s, specification management between 1980s and 2002, guarantee major infrastructure projects between 2002 and 2012, strict restriction from 2012 to 2017, and prohibition basically since 2017" [10]. The reclamation history of port clusters influenced by policy changes in the Bohai Sea could be divided into four stages since 2002, the accelerated period (2002–2007), the peak period (2008–2012), the deceleration period (2013–2017), and the stagnant period (2018–present) (Table 6). The accelerated period occurred during 2002–2007 [49]. In 2005, the central government issued the policy "*Suggestion of the State Council of the People's Republic of China on the Related Issues of Promoting the Development and Opening-up of*

**Table 6. The policy changes of the CLR in China since 1970s.**

| Time | Policy contents |
|------|-----------------|
| 1970s | Completely support for reclamation |
| 1980s | The transition from unpaid to a combination of free and paid |
| 1993 | Strictly control development activities that change the attributes of the sea area or the ecological environments, and the management authority for reclamation sea is not specified. *Provisional Regulations on the Administration of the Use of National Sea Areas* |
| 2002 | Accelerated period (2002~2007). Emphasis on the marine ownership. *Law of the People's Republic of China on the Administration of Sea Area Use* |
| 2006 | *Suggestion of the State Council of the People's Republic of China on the Related Issues of Promoting the Development and Opening-up of the Tianjin New Coastal District.* |
| | *The Shougang Group Relocation from Beijing to Caofeidian* |
| | A national reclamation planning was formulated, and the near-shore waters functional areas were divided into prohibited reclamation areas, restricted reclamation areas, moderate reclamation areas, and reclamation supply areas. |
| 2008 | Peak period (2008~2012). To strengthen the overall planning and guidance of investment projects in marine functional zoning, guarantee the sea demand for major engineering projects, Strengthening the guides of marine function zonation to investment projects and guarantee the demand for sea area of key infrastructure program and improve the special sea area planning and the efficiency of sea area utilization approval work. *Notice on Doing a Good Job of Service Guarantee for Expanding Domestic Demand and Promoting Stable and Fast Economic Development* |
| 2009 | Implementing a reclamation management policy that maintains pressure and leniency to guide and regulate local fixed asset investment and industrial layout. |
| 2010 | The revision of Marine Function Planning and scientifically identify the scale of reclamation sea and the establishment of regional sea utilization planning systems and the management of contiguous reclamation sea area and the implement of annual planning management in the reclaimed area. *Notice on the Enhancement of Reclamation Sea Planning Management.* |
| 2011 | The implementation of mandatory planning management for the annual reclamation scale and clearly reinforcement the whole process supervision of ex ante, interim, and expost of the reclamation project. *Administrative Measures for Reclamation Plannings.* |
| 2012 | The clear requirements on the natural shoreline retention ratio, the ratio of marine ecological red line area. T*he Several Opinions on Establishing the Bohai Sea Marine Ecological Redline System.* |
| 2014 | Deceleration period (2013~2017). The former Ministry of Land and Resources proposed to strengthen land-sea co-ordination, scientifically demarcate and adhere to the marine ecological red line. |
| | The area of small bays and weak ability of self-purification of sea water should be resolutely prohibited reclamation activities. |
| 2015 | Close surveillance for reclamation sea. *Implementation Planning for the Construction of Marine Ecological Civilization of the State Oceanic Administration.* |
| 2017 | Environmental protection inspection and marine inspection should be executed. Reclamation sea should strictly implement the management and control requirements of the ecological protection red line, and scientifically establish a technical system for total amounts control objectives and annual planning index measurement. *Regulations on the Management of Reclamation.* |
| 2017 | The coastline should be included in the management of the marine ecological red line, and the retention ratio of national natural shoreline by 2020 was not less than 35%; *Administrative Measures on Coastline Protection and Utilization.* |
| 2018 | Stagnant period (2018~). Except for major national strategic projects, the general approval of new reclamation projects should be stopped. The government should carry out the restoration and reconstruction of the marine environments in the enclosed area. As for the continuous fragmentation natural wetlands, the restoration and integration remediation should be implemented. *The State Council issued the Notice on Strengthening the Protection of Coastal Wetlands and Strictly Controlling Reclamation.* |
| 2019 | The historical problems of reclamation should be tackled. As for the reclamation sea projects that has acquired the marine utilization rights and don't conduct projects, it should be utilized and to control the area of reclamation to a greatest extent. The reclamation sea projects that have no marine utilization rights, the procedure to use the marine area should be handled based on the restoration schemes drafting of ecological assessment and protection. The illegal marine utilization projects should be punished according to the laws and rules; meanwhile, the ecological restoration should be carried out to conserve the coastal ecosystems as much as possible. *The Ministry of Natural Resources.* |

*the Tianjin Binhai New Area*." The Tianjin Binhai New Area was incorporated into the national overall development strategy in 2005, which accelerated the implementation of CLR [13]. Shougang Group, one of the largest steel group corporations in China, began to move from Beijing to Caofeidian, Hebei Province, owing to land reclamation in 2005. During this period, with the support of the national reclamation policy, the coastal reclamation activities in Tianjin increased rapidly. The reclamation peak period was between 2008 and 2012. The reclamation areas in the Caofeidian Industrial Zone, Huanghua, and Tianjin ports are relatively large. The Caofeidian New Cities, Nanpu Oilfield reclamation area, Beijiang Power Plant (Binhai New Area), and Nangang Industrial Zone are also key reclamation areas. The built-up areas in ports expanded rapidly, and cargo throughput showed vigorous growth. The deceleration period was 2013–2017, with the reclamation areas in the Dalian, Yantai, and Panjin ports becoming relatively large. The built-up areas of ports continued to expand, but the overall area of new reclamation decreased steadily.

Reclamation has been stagnant since 2018 [10]. The Chinese government has begun to implement the most stringent reclamation control, and the State Oceanic Administration will no longer approve general reclamation projects [8]. The Chinese central government has increased its emphasis on the protection of coastal ecosystems and environment and has tightened the approval process for coastal reclamation proposals [10]. The State Council issued the *Notice Strengthening the Protection of Coastal Wetlands and Strictly Controlling Reclamation*. In 2018, the reclamation annual plan index and newly increased reclamation project approval except strategy projects with national significance were comprehensive cancelled [56]. It is apparent that the government, to alleviate the negative impacts on the marine/coastal environment and ecosystem services, is now strengthening the control of reclamation plans [10]. Therefore, previous port reclamation plans may not be implemented well and have caused many ongoing reclamation projects in a stagnation status. Chinese government has legalized and nationalized all lands reclaimed from the sea that have remained idle for a long time [8, 10].

CLR is not allowed in many developed countries; however, it is required in developing countries, especially China [7]. Port reclamation is a practical necessity in coastal ports, where reclamation is the only way to obtain enough water depth for ship transportation. Although the speed of port reclamation has been decreasing, it has not stopped in the past 10 years. The Rotterdam port reclaimed approximately 20 km$^2$ in 2018–2013, and Incheon, Dubai, and Colombo ports have pursued large-scale reclamation. Recently, Nigeria and other developing countries have begun port reclamation to meet their economic development requirements. Moreover, the revision of port plans is driven by ensuring national energy safety, regional industry development, and transportation structure adjustment. Currently, the demand for port-plan revision in coastal areas is vigorous, including the desire for port development toward open sea to avoid the land use contradiction between ports and urban areas. After the prohibition of reclamation, many port master plan schemes that have been approved by the Ministry of Transportation need large-scale revision to seek further environmental assessment approval. Port master plan schemes that have been replied need permission to conduct reclamation activities. When the port plan revision process is finished, the reclamation plan scheme will be cancelled, and port space expansion will be stagnant.

## Conclusion

Coastal zones are a scene of frequent exchange and transformation of materials and energy from climate, hydrology, biology, soil, sediment, and other factors related to both land and sea [57–59]. The coastal region in China covers 13% of the nation's territory, hosts 43.5% of the

nation's population, and contributes 60.8% of the national gross domestic product (GDP) [10, 13, 60]. Land reclamation has been an effective method of alleviating the contradiction between the supply and demand of land and expanding usable land for a variety of purposes [8]. Mainly driven by the rapid economic development and increasing population in coastal areas, high-intensity CLR focusing on the development of port and coastal industries has been regarded as an effective measure to alleviate land shortage [10]. Moreover, reclamation is still growing in China despite being widely criticized for its negative impacts, and it often incurs relatively low costs but yields huge profits. In this study, remote sensing images, field investigations, statistical data, and associated plan data were used to reveal changes in coastline structures, the reclamation process of port clusters, and the land use structures in port reclamation areas since 2002. Findings showed that the area of reclamation of the 13 ports during 2002–2018 was 2,300 km$^2$. The natural coastline length in Tianjin decreased by 47.5 km, whereas the artificial coastline length increased by 46.6 km. However, the ratio of built-up areas within the ports was found to be only 32.5%, and approximately 48.3% of the reclaimed areas have no construction projects. This paper also documents the existing legislation and regulations related to CLR and demonstrates the need for regulation and careful planning to propose suggestions for the reclamation of port clusters in the Bohai Sea, China. This study aims not to defend reclamation activities but to identify the main land use types within reclaimed land to better conduct reclamation and restoration projects that are ecological and environmentally friendly. Future port reclamation should not be totally prohibited; it needs fine management based on optimization of the current ports' reclaimed areas. This is the innovation of this study.

There are several limitations associated with this research. Remote sensing images from 2002 and 2018 were used to identify the coastline lengths and port reclamation percentages. No continuous time series data were used to explain the annual changes, which may have led to some inaccuracies. Regarding data quality, the 30 m resolution led to difficulties in identifying the detailed inner structures of ports; images that are more precise are required to clarify land use structures. Moreover, the future trend (2030) was analyzed the port reclamation trend, but port reclamation is impacted by policy changes substantially. Future research should thus focus on the ecological effects of port reclamation and perform an integrated assessment of port risks resulting from port reclamation under climate change.

## Acknowledgments

We are thankful to the Academic Editor and the two anonymous reviewers for their constructive comments that have greatly improved the quality of this research. Jie Liu, Li Lei at Transport Planning and Research Institute, Ministry of Transport of the People's Republic of China also provided useful comments during field survey and research implementation.

## Author Contributions

**Conceptualization:** Gaoru Zhu.

**Data curation:** Minxuan Liang, Liguo Zhang.

**Funding acquisition:** Gaoru Zhu.

**Investigation:** Gaoru Zhu, Zhenglei Xie.

**Methodology:** Liguo Zhang.

**Project administration:** Honglei Xu.

**Resources:** Honglei Xu.

**Software:** Jinxiang Cheng.

**Supervision:** Honglei Xu.

**Validation:** Minxuan Liang.

**Visualization:** Jinxiang Cheng, Yujian Gao.

**Writing – original draft:** Zhenglei Xie.

**Writing – review & editing:** Zhenglei Xie.

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
