## [Decision Letter · Decision Letter 0]

9 Aug 2021

PONE-D-21-22241

Land reclamation pattern and environmental regulation guidelines for port clusters in the Bohai Sea, China

PLOS ONE

Dear Dr. Xie,

Thank you for submitting your manuscript to PLOS ONE. After careful consideration, we feel that it has merit but does not fully meet PLOS ONE’s publication criteria as it currently stands. Therefore, we invite you to submit a revised version of the manuscript that addresses the points raised during the review process.

We look forward to receiving your revised manuscript.

Kind regards,

Jun Yang

Academic Editor

PLOS ONE

Journal Requirements:

3. We note that Figures 1, 3, 4 and 5  in your submission contain map/satellite images which may be copyrighted. All PLOS content is published under the Creative Commons Attribution License (CC BY 4.0), which means that the manuscript, images, and Supporting Information files will be freely available online, and any third party is permitted to access, download, copy, distribute, and use these materials in any way, even commercially, with proper attribution. For these reasons, we cannot publish previously copyrighted maps or satellite images created using proprietary data, such as Google software (Google Maps, Street View, and Earth). For more information, see our copyright guidelines: http://journals.plos.org/plosone/s/licenses-and-copyright.

a) You may seek permission from the original copyright holder of Figures 1, 3, 4 and 5 to publish the content specifically under the CC BY 4.0 license.  

4. We note that Figure 2 in your submission contain copyrighted images. All PLOS content is published under the Creative Commons Attribution License (CC BY 4.0), which means that the manuscript, images, and Supporting Information files will be freely available online, and any third party is permitted to access, download, copy, distribute, and use these materials in any way, even commercially, with proper attribution. For more information, see our copyright guidelines: http://journals.plos.org/plosone/s/licenses-and-copyright.

a) You may seek permission from the original copyright holder of Figure 2 to publish the content specifically under the CC BY 4.0 license. 

Reviewers' comments:

Reviewer's Responses to Questions

**Comments to the Author**

1. Is the manuscript technically sound, and do the data support the conclusions?

Reviewer #1: Yes

Reviewer #2: Yes

2. Has the statistical analysis been performed appropriately and rigorously? 

Reviewer #1: Yes

Reviewer #2: Yes

3. Have the authors made all data underlying the findings in their manuscript fully available?

Reviewer #1: Yes

Reviewer #2: Yes

4. Is the manuscript presented in an intelligible fashion and written in standard English?

Reviewer #1: Yes

Reviewer #2: Yes

5. Review Comments to the Author

Reviewer #1: This paper tries to identify and solve in important question. The methodology is sound and the data seems valid. However, the paper provides too many pieces of information without effectively provide a reasonable link among them and explain how that information could be used in answering the research question and for better land reclamation and environmental regulation for the future of the study region. I have some major concerns listed below:

Information on funding and competing Interests should be placed at the end of the paper before reference list.

Introduction: very poor with limited literature about port cluster, coastal land reclamation, evolution stages, pattern optimization etc.

Study area and research method: very weak! How come we get to Table 1? And what is the justification of adopting equation 1-2? Source?

Results：still simply copy paste from commissioned works.

Discussion: some interesting insights but still lack of coherency with the introduction/literature.

The current section is not directly related to the analysis of this article.

Conclusion and policy implications: it mostly looks like a summary of the work done and the results obtained. Also, there should be some content in the conclusion regarding the limitations of the current research and future work possibilities.

There are more language and grammar issues than mentioned above. I highly recommend using a professional editor to correct them before re-submission.

Some literature to consult:

Demand prediction and regulation zoning of urban-industrial land: Evidence from Beijing-Tianjin-Hebei Urban Agglomeration, China. Environ Monit Assess 191, 412 (2019). https://doi.org/10.1007/s10661-019-7547-4

Understanding land surface temperature impact factors based on local climate zones, Sustainable Cities and Society (2021), doi: https://doi.org/10.1016/j.scs.2021.102818

Coupling Coordination Relationships between Urban-industrial Land Use Efficiency and Accessibility of Highway Networks: Evidence from Beijing-Tianjin-Hebei Urban Agglomeration, China. Sustainability. 2019; 11(5):1446.

Reviewer #2: After a very careful reading of the work entitled "Land reclamation pattern and environmental regulation guidelines for port clusters in the Bohai Sea, China", I have found a very well-done work, well presented and organized, clear in concepts and methodology. The topic and context attract attention for many readers from various disciplines. The study is worth to be published in PLOS ONE after conducting the revisions.

1. Please further elaborate the innovation of the paper.

2. The language of this manuscript needs to be revised by native English speaking experts.

6. PLOS authors have the option to publish the peer review history of their article (what does this mean?). If published, this will include your full peer review and any attached files.

Reviewer #1: No

Reviewer #2: No

---

## [Author Response · Author response to Decision Letter 0]

29 Aug 2021

Responses to Editor-in Chief’s recommendation

(1) Question: Thank you for submitting your manuscript to PLoS ONE. After careful consideration, we feel that it has merit but does not fully meet PLOS ONE’s publication criteria as it currently stands. Therefore, we invite you to submit a revised version of the manuscript that addresses the points raised during the review process.

Response: We are thankful to the editors and reviewers for giving constructive comments on the manuscript and providing a merit evaluation and a revision opportunity. The editors believe that the current version does not fully meet PLoS ONE's publication criteria. These constructive comments proposed by the reviewers are helpful to the improvement of the previous version. We have identified the reviewers' points one-by-one raised during the review process carefully according to the reviewer’s comments and we have submitted a revised manuscript. We hope that the revised manuscript would be accepted very soon. 

 (2) Question: Please submit your revised manuscript by Sep 23 2021 11:59PM. If you will need more time than this to complete your revisions, please reply to this message or contact the journal office at plosone@plos.org. Please include the following items when submitting your revised manuscript:

Response: We are thankful for you providing a revision opportunity to our manuscripts and we will resubmit the revised manuscript before the due time based on the above website. The file labeled 'Response to Reviewers' has been uploaded as separate file and has been responded to each point raised by the academic editor and reviewers. Moreover, we have also submitted a separate file labeled "Revised Manuscript with Track Changes", which has highlighted changes made to the original version. The unmarked version of revised paper without tracked changes has been also uploaded as separated file. Please refer to the revised manuscript.

We are thankful that the EIC and the reviewers have requested a number of additions and clarification, particularly regarding improvements to the presentation of the work and discussion of related literature. We have revised the manuscript carefully and addressed the comments raised by the reviewers. Meanwhile, we also provided a suitable rebuttal to each reviewer comment not addressed.

(3) Question: Response: We are thankful to your comments and there are changes for financial disclosure. Funding projects has been listed in the revised manuscript and cover letter. We have carefully read the guidelines for resubmitting the figure files and believe that the figure file format should meet the requirements of the journal. 

(4) Question: If applicable, we recommend that you deposit your laboratory protocols in protocols.io to enhance the reproducibility of your results. Protocols.io assigns your protocol its own identifier (DOI) so that it can be cited independently in the future. For instructions see: http://journals.plos.org/plosone/s/submission-guidelines#loc-laboratory-protocols. Additionally, PLOS ONE offers an option for publishing peer-reviewed Lab Protocol articles, which describe protocols hosted on protocols.io. Read more information on sharing protocols at https://plos.org/protocols?utm_medium=editorial-email&utm_source=authorletters&utm_campaign=protocols.

Response: We are thankful to the editor comments. The revised manuscript focuses the reclamation area of the 13 ports during 2002–2018, changes of coastline length, and reclamation stages. The findings have vital implications for scientifically regulating the spatial pattern and exploring the utility of port reclamation, and promoting regional sustainable development of coastal zones in dense port distribution. The revised manuscript has no laboratory analysis process and therefore, there is no laboratory protocols. Moreover, we have read the Submission Guidelines carefully in https://journals.plos.org/plosone/s/submission-guidelines#loc-laboratory-protocols. Thank you very much. 

(5) Question: Journal Requirements:

Response: We have read these additional requirements carefully and believe the revised manuscript should meet the additional requirements. Thank you very much. 

(6) Question: We note that you have stated that you will provide repository information for your data at acceptance. Should your manuscript be accepted for publication, we will hold it until you provide the relevant accession numbers or DOIs necessary to access your data. If you wish to make changes to your Data Availability statement, please describe these changes in your cover letter and we will update your Data Availability statement to reflect the information you provide.

Response: We believe all the data demonstrated in the revised manuscript has been listed and the readers would access the data without any restriction. 

(7) Question: We note that Figures 1, 3, 4 and 5 in your submission contain map/satellite images which may be copyrighted. All PLOS content is published under the Creative Commons Attribution License (CC BY 4.0), which means that the manuscript, images, and Supporting Information files will be freely available online, and any third party is permitted to access, download, copy, distribute, and use these materials in any way, even commercially, with proper attribution. For these reasons, we cannot publish previously copyrighted maps or satellite images created using proprietary data, such as Google software (Google Maps, Street View, and Earth). For more information, see our copyright guidelines: http://journals.plos.org/plosone/s/licenses-and-copyright.

Response: We are thankful to the editor’s comments. The satellite images of Figure 1 were free to download from the website (http://glovis.usgs.gov/) and is freely available online. The Figure 1 was not produced from Google Maps and it was produced from the downloaded satellite. These satellite images are all open to the public on line and is allowed to access, download, copy, distribute in any way. Therefore, we believe Figures 1 has not copyrighted maps.

(8) Question: You may seek permission from the original copyright holder of Figures 1, 3, 4 and 5 to publish the content specifically under the CC BY 4.0 license. 

 Response: We are thankful to your constructive comments. We have contacted the original copyright holders to seek the allow to use the materials. These figures are freely distributed in website and we believe the Figures in the revised manuscript should meet the requirements of the journals. 

(9) Question: If you are unable to obtain permission from the original copyright holder to publish these figures under the CC BY 4.0 license or if the copyright holder’s requirements are incompatible with the CC BY 4.0 license, please either i) remove the figure or ii) supply a replacement figure that complies with the CC BY 4.0 license. Please check copyright information on all replacement figures and update the figure caption with source information. If applicable, please specify in the figure caption text when a figure is similar but not identical to the original image and is therefore for illustrative purposes only. The following resources for replacing copyrighted map figures may be helpful: USGS National Map Viewer (public domain): http://viewer.nationalmap.gov/viewer/ The Gateway to Astronaut Photography of Earth (public domain): http://eol.jsc.nasa.gov/sseop/clickmap/ Maps at the CIA (public domain): https://www.cia.gov/library/publications/the-world-factbook/index.html and https://www.cia.gov/library/publications/cia-maps-publications/index.html

Landsat: http://landsat.visibleearth.nasa.gov/ USGS EROS (Earth Resources Observatory and Science (EROS) Center) (public domain): http://eros.usgs.gov/#

Response: We are thankful to your constructive comments. We have known that you provided the following resources for replacing copyrighted map figures complies with the CC BY 4.0 license. Our figures are download from the website of USGS website and we believe it should meet your requirements.

(10) Question: We note that Figure 2 in your submission contain copyrighted images. All PLOS content is published under the Creative Commons Attribution License (CC BY 4.0), which means that the manuscript, images, and Supporting Information files will be freely available online, and any third party is permitted to access, download, copy, distribute, and use these materials in any way, even commercially, with proper attribution. For more information, see our copyright guidelines: http://journals.plos.org/plosone/s/licenses-and-copyright.

Response: The Figure 2 is taken from the corresponding author and we believe we have the copyright. 

 (11) Question: You may seek permission from the original copyright holder of Figure 2 to publish the content specifically under the CC BY 4.0 license. 

If you are unable to obtain permission from the original copyright holder to publish these figures under the CC BY 4.0 license or if the copyright holder’s requirements are incompatible with the CC BY 4.0 license, please either i) remove the figure or ii) supply a replacement figure that complies with the CC BY 4.0 license. Please check copyright information on all replacement figures and update the figure caption with source information. If applicable, please specify in the figure caption text when a figure is similar but not identical to the original image and is therefore for illustrative purposes only.

Response: The copyright of Figure 2 belongs to us and it was taken at July, 2019. At that time, the first author and the corresponding author conducted a field investigation of the landscape structure within the ports at the Bohai Rim

 Comments to the Author

(1) Question: Is the manuscript technically sound, and do the data support the conclusions? The manuscript must describe a technically sound piece of scientific research with data that supports the conclusions. Experiments must have been conducted rigorously, with appropriate controls, replication, and sample sizes. The conclusions must be drawn appropriately based on the data presented.

Reviewer #1: Yes

Reviewer #2: Yes

Response: Thank the reviewer’s high evaluation on the manuscript. We appreciate the reviewers' assessment to our manuscript. The reviewer thought that the manuscript is technically sound and the data support the conclusion. 

 Coastal land reclamation (CLR) has become a frequent approach to alleviate land shortages with the reclamation of port areas being the most prominent. The Bohai Sea is the most concentrated area of port reclamation worldwide. The manuscript applied the remote sensing images and field investigation to address the changes in different type coastlines and the CLR process of these ports in the reclaimed areas of the Bohai Sea. The port land reclamation experienced periods of acceleration, peak, deceleration, and stagnation since 2002, and the planned schedule of reclamation should be constrained. The findings have vital implications for scientifically regulating the spatial pattern and exploring the utility of port reclamation, and promoting regional sustainable development of coastal zones in dense port distribution. We believe that the revised manuscript has broad readers all over the world. The conclusion has been drawn appropriately based on the data presented. 

 (2) Question: Has the statistical analysis been performed appropriately and rigorously?

Reviewer #1: Yes

Reviewer #2: Yes

Response: Thank the reviewer’s evaluation on the manuscript. The manuscript has applied statistical analysis appropriately. We think that the current statements of the revised manuscript will meet the standard that the journal demands and the readers have no difficulties in understanding the content of the manuscript. We express our gratitude's to the reviewer for the helpful comments.

 (3) Question: Have the authors made all data underlying the findings in their manuscript fully available?

Reviewer #1: Yes

Reviewer #2: Yes

Response: Thank the reviewer’s evaluation on the manuscript. We have known that the PLoS ONE Data policy requires the authors to make all data underlying the findings described in their manuscript fully available without restriction. The data presented in the manuscript has been provided as part of the manuscript. We express our gratitude's to the reviewer for the helpful comments.

 (4) Question: Is the manuscript presented in an intelligible fashion and written in standard English?

Reviewer #1: Yes

Reviewer #2: Yes

Response: Thank the reviewer’s evaluation on the manuscript. The manuscript is presented in an intelligible fashion and written in standard English. More important, the writing expression has been revised by a Native English Speaker carefully from Elsevier. There is no any typographical or grammatically errors. Therefore, we believe the language in submitted article is clear, correct, and unambiguous. Please refer to the revised manuscript. Thank you. 

(5) Question: Review Comments to the Author. 

Reviewer #1: 

(1) Question: This paper tries to identify and solve in important question. The methodology is sound and the data seems valid. However, the paper provides too many pieces of information without effectively provide a reasonable link among them and explain how that information could be used in answering the research question and for better land reclamation and environmental regulation for the future of the study region. I have some major concerns listed below:

Information on funding and competing Interests should be placed at the end of the paper before reference list.

Response: Thank the reviewer’s evaluation on the manuscript. The reviewers think that the coastal port reclamation is an important question and the author tries to solve the crucial problem. Port reclamation is the main types of coastal reclamation. However, the spatial extent, percentages, and processes of these newly reclaimed ports are largely unknown. The reviewer think that the methodology is sound and the data seems valid. Thank the reviewer’s evaluation on the manuscript. 

Moreover, we should not blame the port reclamation for several reasons. First, not all the reclaimed area is used as port construction and only 26.3% of reclaimed areas were applied for port construction in Tianjin and Tangshan ports.

Regarding with the link among the information, we have revised the manuscript and provided the research question for better land reclamation and environmental regulation for the future of the study region. 

 (2) Question: Introduction: very poor with limited literature about port cluster, coastal land reclamation, evolution stages, pattern optimization etc.

Response: Thank the reviewer’s evaluation on the manuscript. We have revised the manuscript in the Introduction and added some literature about port cluster, coastal land reclamation, evolution stages, pattern optimization. Please refer to the revised manuscript. Thank you again. 

(3) Question: Study area and research method: very weak! How come we get to Table 1? And what is the justification of adopting equation 1-2? Source?

Response: We are thankful to the reviewer’s comments. The section study area and research method have been improved based on the reviewer’s comments. In the current version, some detailed information about study area have been added and the source of equation 1-2 have also been provided. Please refer to the revised manuscript. 

(4) Question: Results: still simply copy paste from commissioned works.

Response: We are thankful to the reviewer’s comments. The Results section has been modified according to the reviewer’s comments. 

(5) Question: Discussion: some interesting insights but still lack of coherency with the introduction/literature. The current section is not directly related to the analysis of this article.

Response: We are thankful to the reviewer’s comments. The reviewer thinks that the discussion has some interesting insight. As for the coherency with the introduction/literature, we have modified the discussion section and related information has been added in the revised manuscript. 

(6) Question: Conclusion and policy implications: it mostly looks like a summary of the work done and the results obtained. Also, there should be some content in the conclusion regarding the limitations of the current research and future work possibilities.

Response: We are thankful to the reviewer’s comments. In the Conclusion and Discussion, we have added many sentences to illustrate the limitation of the current research and future work. We believe the revised conclusion should meet the requirements of the reviewers. Please refer to it. 

(7) Question: There are more language and grammar issues than mentioned above. I highly recommend using a professional editor to correct them before re-submission.

Response: We are thankful to the reviewer’s comments. More important, the writing expression has been revised by a Native English Speaker carefully from Elsevier. There is no any typographical or grammatically errors. Therefore, we believe the language in submitted article is clear, correct, and unambiguous. Please refer to the revised manuscript. 

(8) Question: Some literature to consult:

Demand prediction and regulation zoning of urban-industrial land: Evidence from Beijing-Tianjin-Hebei Urban Agglomeration, China. Environ Monit Assess 191, 412 (2019). https://doi.org/10.1007/s10661-019-7547-4

Understanding land surface temperature impact factors based on local climate zones, Sustainable Cities and Society (2021), doi: https://doi.org/10.1016/j.scs.2021.102818

Coupling Coordination Relationships between Urban-industrial Land Use Efficiency and Accessibility of Highway Networks: Evidence from Beijing-Tianjin-Hebei Urban Agglomeration, China. Sustainability. 2019; 11(5):1446.

Response: We are thankful to the reviewer’ comments. The three published articles have been added in the reference list. Please refer to it. 

Reviewer #2: 

(1) Question: After a very careful reading of the work entitled "Land reclamation pattern and environmental regulation guidelines for port clusters in the Bohai Sea, China", I have found a very well-done work, well presented and organized, clear in concepts and methodology. The topic and context attract attention for many readers from various disciplines. The study is worth to be published in PLOS ONE after conducting the revisions.

Response: We are thankful to the reviewer’s To Be Published evaluation on the manuscript. The reviewer thinks that it is a well-done work, well presented and organized, clear in concepts and methodology. Meanwhile, the review believe the topic and context attract attention for many readers from various disciples. The manuscript takes the port reclamation in the Bohai Sea as an example to demonstrate the coastline change, percentage of port reclamation in all the reclaimed area. The manuscript is not to encourage the reclamation in coastal area, but to identify the truth of port reclamation. It has important implication for decision-makers to adopt reasonable reclamation policy in future. We express our gratitude to the kind reviewer for his constructive comments. Meanwhile, the reviewer has provided some comments to improve the quality of this manuscript.

(2) Question: Please further elaborate the innovation of the paper.

The language of this manuscript needs to be revised by native English speaking experts.

Response: We are thankful to the reviewer’s comments. The innovation of the paper has been proposed in the Conclusion and Abstract section. Moreover, the manuscript has been polished by a native English speaking expert. We believe the manuscript has been improved much in language.

---

## [Decision Letter · Decision Letter 1]

20 Sep 2021

PONE-D-21-22241R1Land reclamation pattern and environmental regulation guidelines for port clusters in the Bohai Sea, ChinaPLOS ONE

Dear Dr. Xie,

Thank you for submitting your manuscript to PLOS ONE. After careful consideration, we feel that it has merit but does not fully meet PLOS ONE’s publication criteria as it currently stands. Therefore, we invite you to submit a revised version of the manuscript that addresses the points raised during the review process.

We look forward to receiving your revised manuscript.

Kind regards,

Jun Yang

Academic Editor

PLOS ONE

Journal Requirements:

Additional Editor Comments (if provided):

1. The layout of the paper is very bad, especially the tables and headings.

2. I suggest the results should be better discussed and justified, such as whether they are consistent with previous studies or analyzing the reasons for the empirical results.At present, there are many contents in the discussion section that are not closely related to this paper.

Reviewers' comments:

Reviewer's Responses to Questions

**Comments to the Author**

1. If the authors have adequately addressed your comments raised in a previous round of review and you feel that this manuscript is now acceptable for publication, you may indicate that here to bypass the “Comments to the Author” section, enter your conflict of interest statement in the “Confidential to Editor” section, and submit your "Accept" recommendation.

Reviewer #1: (No Response)

Reviewer #2: All comments have been addressed

2. Is the manuscript technically sound, and do the data support the conclusions?

Reviewer #1: (No Response)

Reviewer #2: Yes

3. Has the statistical analysis been performed appropriately and rigorously? 

Reviewer #1: (No Response)

Reviewer #2: Yes

4. Have the authors made all data underlying the findings in their manuscript fully available?

Reviewer #1: (No Response)

Reviewer #2: Yes

5. Is the manuscript presented in an intelligible fashion and written in standard English?

Reviewer #1: (No Response)

Reviewer #2: Yes

6. Review Comments to the Author

Reviewer #1: Although the author has done a lot of revision work, there are still many problems and I cannot recommend it for publication at the present version.

1. The layout of the paper is very bad, especially the tables and headings.

2. I suggest the results should be better discussed and justified, such as whether they are consistent with previous studies or analyzing the reasons for the empirical results.At present, there are many contents in the discussion section that are not closely related to this paper.

Reviewer #2: This manuscript has been revised in detail according to the comments of the reviewers. The current version is worth publishing.

7. PLOS authors have the option to publish the peer review history of their article (what does this mean?). If published, this will include your full peer review and any attached files.

Reviewer #1: No

Reviewer #2: No

---

## [Author Response · Author response to Decision Letter 1]

11 Oct 2021

Responses to Editor-in Chief’s recommendation

(1) Question: Thank you for submitting your manuscript to PLOS ONE. After careful consideration, we feel that it has merit but does not fully meet PLOS ONE’s publication criteria as it currently stands. Therefore, we invite you to submit a revised version of the manuscript that addresses the points raised during the review process.

Response: We express our gratitude to you and the two anonymous reviewers for providing helpful suggestion in term of manuscript structure toward the revised manuscript. The Editor-in-Chief has made a decision of merit evaluation and provided a revision opportunity for the manuscript. The Editor-in-Chief maintain that the current status of the manuscript does not fully meet PLoS ONE's publication standard. These useful comments brought by the two reviewers are totally positive effective to the improvement of the revision version. We have addressed the reviewers' points one-by-one carefully based on the reviewer’s comments and we have submitted the second revised manuscript before the due date. We wish that the second revised manuscript would be accepted in the journal immediately. 

 (2) Question: Please submit your revised manuscript by Nov 04 2021 11:59PM. If you will need more time than this to complete your revisions, please reply to this message or contact the journal office at plosone@plos.org. Please include the following items when submitting your revised manuscript:

Response: Very appreciated for your decision. The first author and the corresponding author will resubmit the second revised manuscript. The file that has been labeled as 'Response to Reviewers' has been prepared as a separate file and has been tackled to each inquiry proposed by the academic editor and the two anonymous reviewers. Additionally, we have prepared a separate file labeled "Revised Manuscript with Track Changes", which has identified the changes made to the original version. The unmarked changes have been also uploaded as a separated file. Please read the current revised manuscript.

We thank the EIC and the two reviewers for proposing a number of additions and clarification, particularly toward the improvements to the discussion of related literature. We have addressed the manuscript thoroughly and addressed the comments raised by the reviewers. At the same time, we also attached a rigorous rebuttal to each comment that has not been addressed by the two reviewers.

(3) Question: Response: We are thankful to your constructive suggestion and there are no changes for financial disclosure. Because the funding information have been list in the previous revised manuscript and cover letter. We have carefully read the guidelines and we feel very strong that the format of the second revised manuscript should meet the criteria of the journal. 

(4) Question: If applicable, we recommend that you deposit your laboratory protocols in protocols.io to enhance the reproducibility of your results. Protocols.io assigns your protocol its own identifier (DOI) so that it can be cited independently in the future. For instructions see: http://journals.plos.org/plosone/s/submission-guidelines#loc-laboratory-protocols. Additionally, PLOS ONE offers an option for publishing peer-reviewed Lab Protocol articles, which describe protocols hosted on protocols.io. Read more information on sharing protocols at https://plos.org/protocols?utm_medium=editorial-email&utm_source=authorletters&utm_campaign=protocols.

Response: We express our gratitude to the two anonymous reviewers for their constructive suggestions. The second version of revised manuscript highlights the reclamation status and reclamation stages, i.e., in area changes, especially of the coastline changes of the 13 ports during 2002–2018. The obtained results have important meanings for scientifically regulating the spatial pattern in coastal area, and proposing next reclamation policy in dense port distribution in the Bohai Rim area. The revised version has no laboratory analysis process and therefore, there is no laboratory protocols. Moreover, we have read the Submission Guidelines carefully in https://journals.plos.org/plosone/s/submission-guidelines#loc-laboratory-protocols. Thank you very much. 

(5) Question: Please review your reference list to ensure that it is complete and correct. If you have cited papers that have been retracted, please include the rationale for doing so in the manuscript text, or remove these references and replace them with relevant current references. Any changes to the reference list should be mentioned in the rebuttal letter that accompanies your revised manuscript. If you need to cite a retracted article, indicate the article’s retracted status in the References list and also include a citation and full reference for the retraction notice.

Response: We are thankful to the EIC’s suggestion for the reference list. We have checked the reference list and believe the current reference list is complete and correct. If the reference has been changes, such as delete or replace, we will identify it in the rebuttal letter and the revised manuscript. 

References

1. Meng WQ, Hu BB, He MX, Liu BQ, Mo XQ, Li HY, et al. Temporal-spatial variations and driving factors analysis of coastal reclamation in China. Estuar Coast Shelf Sci. 2017; 191: 39–49.

2. Yang J, Ren JY, Sun DQ, Xiao XM, Xia JH, Jin C, et al. Understanding land surface temperature impact factors based on local climate zones. Sustain Cities Soc. 2021; 69:102818. https://doi.org/10.1016/j.scs.2021.102818.

3. Zuo TL, Zha YP, Nei XJ, Li R, Qi Y, Dong M. Research on planning concept of ecological port. Port Waterw. Eng. 2017; 5: 56–61 (in Chinese).

4. Izaguirre C, Losada IJ, Camus P, Vigh JL, Stenek V. Climate change risk to global port operations. Nat Clim Change. 2021; 11: 14–20. https://doi.org/10.1038/s41558-020-00937-z.

5. Martínantón M, Negro V, Campo JMD, Lópezgutiérrez JS, Esteban MD. Review of coastal land reclamation situation in the world. J. Coast Res SI. 2016; (75): 667–671.

6. Hanson SE, Nicholls RJ. Demand for ports to 2050: climate policy, growing trade and the impacts of sea-level rise. Earth’s Future. 2020; http://doi.org/10.1029/2020EF001543.

7. Chen YY. Survey on influence of maritime port cluster effect on offshore regional economy based on grey correlation model. J. Coast Res 2019; 94: 707-711. Doi:10.2112/SI94–140.1.

8. Chen YP, Wei YQ, Peng LH. Ecological technology model and path of seaport reclamation construction. Ocean. Coastal. Manage. 2018; 165: 244–257. https://doi.org/10.1016/j.ocecoaman.2018.08.031.

9. Pang H, Dong SH. Inventory collaboration in coastal cluster supply chain. J of Coastal Res. 2019; 94: 617–620. 

10. Wang W, Liu H, Li YQ, Su JL. Development and management of land reclamation in China. Ocean Coastal Manage. 2014; 102: 415–425.

11. Xu JY, Zhang ZX, Zhao XL, Wen QK, Zuo LJ, Wang X, et al. Spatial-temporal analysis of coastline changes in northern China from 2000 to 2012. Acta Geograph. Sin. 2013; 68(5): 651–660.

12. Sengupta D, Chen RS, Meadows ME. Building beyond land: an overview of coastal land reclamation in 16 global megacities. Appl Goegr. 2018; 90: 229–238.

13. Chen WG, Wang DC, Huang Y, Chen LD, Zhang LH, Wei XW, et al. Monitoring and analysis of coastal reclamation from 1995–2015 in Tianjin Binhai New Area, China. Sci Rep. 2017; 7: 3850.

14. Tian B, Wu WT, Yang ZQ, Zhou YX. Drivers, trends, and potential impacts of long-term coastal reclamation in China from 1985 to 2010. Estuar Coast Shelf Sci. 2016; 170: 83–90.

15. Wahl T, Haigh ID, Nicholls RJ, Arns A, Dangendorf S, Hinkel J, et al. Understanding extreme sea levels for broad-scale coastal impact and adaptation analysis. Nature Communication. 2017; 8:16075.

16. Li CX, Gao X, He BJ, Wu JY, Wu KN. 2019. Coupling coordination relationships between urban-industrial land use efficiency and accessibility of highway networks: evidence from Beijing-Tianjin-Hebei Urban agglomeration, China. Sustainability. 11: 1446, doi:10.3390/su11051446.

17. Li JG, Pu LJ, Zhu M, Zhang J, Li P, Dai XQ, et al. Evolution of soil properties following reclamation in coastal areas: a review. Geoderma. 2014; 226-227(1): 130–139.

18. Piersma T. Threats to intertidal soft-sediment ecosystems. University of Groningen. 2009

19. Jaramillo E, Lagos NA, Labra FA, Paredes E, Acuna E, Melnic D, et al. Recovery of black-necked swans, macrophytes and water quality in a Ramsar wetland of southern Chile: assessing resilience following sudden anthropogenic disturbances. Sci. Total Environ. 2018; 628–629, 291–301.

20. UNEP, 2016. Population within 100 Kilometers of Coast. Available at http://geodata.grid.unep.ch/options.phs?selectedID=280&selectedDatasettype=1. (Accessed 16 May 2016)

21. Zhu GR, Xie ZL, Xu XG. The landscape change and theory of orderly reclamation sea based on coastal management in rapid industrialization area in Bohai Bay, China. Ocean Coastal Manage. 2016; 133, 128–137.

22. Chee SY, Othman AG, Sim YK, Adam ANM, Firth LB. Land reclamation and artificial islands: walking the tightrope between development and conservation. Glob. Ecol Conserv. 2017; 12: 80–95.

23. Kim RH, Kim JH, Ryu JS, Koh DC. Hydrogeochemical characteristics of groundwater influenced by reclamation, seawater intrusion, and land use in the coastal area of Yeonggwang, Korea. Geosci J. 2019; 23: 603–619. 

24. Lu QQ, Bai JH, Zhang GL, Wu JJ. Effects of coastal reclamation history on heavy metals in different wetland type soils in the Pearl River Delta: levels, sources and ecological risks. J of Clean Prod. 2020; 272: 122668. 

26. Bulleri F, Chapman MG. The introduction of coastal infrastructure as a driver of change in marine environments. J Appl Ecol. 2010; 47(1): 26–35.

27. Guneroglu N, Acar C, Guneroglu A, Dihkan M. Coastal land degradation and character assessment of Southern Black Sea landscape. Ocean. Coastal Manag. 2015; 118: 282–289.

28. Jongman B, Ward PJ, Aerts JC. Global exposure to river and coastal flooding: long term trends and changes. Global Environment Change. 2012; 22(4): 823–835.

29. Cheng ZX, Jalon-Rójas I, Wang XH, Liu Y. Impacts of land reclamation on sediment transport and sedimentary environment in a macro-tidal estuary. Estuar Coast Shelf Sci. 2020; 242: 106861.

30. Wu WT, Yang ZQ, Tian B, Huang Y, Zhou YX, Zhang T. Impacts of coastal reclamation on wetlands: loss, resilience, and sustainable management. Estuar. Coast Shelf Sci. 2018; 210:153–161.

31. Xu Y, Cai YP, Sun T, Tan Q. A multi-scale integrated modeling framework to measure comprehensive impact of coastal reclamation activities in Yellow River estuary, China. Mar Pollut Bull. 2017; 122(1-2): 27–37.

32. Suo AN, Wang P, Yuan DW, Yu YH, Zhang MH. Study on monitoring and analysis of existing sea reclamation resource based on high resolution satellite remote sensing imagery- a case in south coast of Yingkou. Acta Oceanol Sin. 2016; 38(9): 54-63 (in Chinese).

33. Peng B, Xie H, Xie Z, Li XL, Yang RG. Sea reclamation dynamic change and driving factors analysis within a port. Geospatial Infor. 2019; 3: 109–112.

34. Wei F, Han GX, Han M, Zhang JP, Li YZ, Zhao J. Temporal-spatial dynamic evolution and mechanism of shoreline and the sea reclamation in the Bohai Rim during 1980-2017. Scien Geogr Sin. 2019; 39(6), 997–1007.

35. Yin YH. Thinking on large area reclamation in Caofeidian, Tangshan city, Hebei province. Mar Geol Lett. 2007; 3: 1–10 (in Chinese).

36. Duan HB, Zhang H, Huang QF, Zhang YK, Hu MW, Niu YN, et al. Characterization and environmental impact analysis of sea land reclamation activities in China. Ocean Coast Manage. 2016; 130: 128–137. 

37. Pelling HE, Uehara K, Green JAM. The impact of rapid coastline changes and sea level rise on the tides in the Bohai Sea, China. J. Geophys. Res: Oceans. 2013; 118(7): 3462–3472.

38. Shen CC, Shi HH, Zheng W, Li F, Peng ST, Ding DW. Study on the cumulative impact of reclamation activities on ecosystem health in coastal waters. Mar Pollut Bull. 2016; 103: 144–150.

39. Fu YB, Cao K, Wang F, Zhang FS. Research of quantitative assessment method of reclamation intensity and potentiality. Ocean Development and Management. 2010. 27(1): 27–30.

40. Pang RZ. Dynamic evaluation of main sea ports in mainland China based on DEA model. Economic Research Journal. 2006.(6): 92–100.

41. Xu N. Research on spatial and temporal variation of China mainland coastline and coastal engineering. University of Chinese Academy of Sciences. 2016.

42. Li S, Zhao S, Zhou X, Li H. Analysis on environment of Changlu salt pan area in North China. Tianjin Sci. Technol. 2011, 5: 114–116 (in Chinese).

43. Jin YD, Zhang QF, Li XB, Wang LN, Ye FJ. Effects of Tianjin reclamation projects on the Bohai Bay water exchange. Mar. Sci. Bull. 2017, 36(5): 578–584.

44. Xiao QC, Wei YS, Wang YW. Driving factors of coastal wetland degradation in Binhai New Area of Tianjin. Acta Sci. Circumst. 2012; 2: 480–488 (in Chinese).

45. Yue Q, Zhao M, Yu HM, Xu W, Qu L. Total quantity control and intensive management system for reclamation in China. Ocean Coast Manage. 2016, 120, 64–69.

46. Yue Q, Zhao M, Yu HM, Xu W, Ou L. Total quantity control and intensive management system for reclamation in China. Ocean Coast Manag. 2016; 120: 64–69. http://dx.doi.org/10.1016/j.ocecoaman.2015.11.026.

47. Hearly MG, Hickey KR. Historic land reclamation in the intertidal wetlands of the Shannon estuary, western Ireland. J Coast. Res. 2002, S36: 365–373. 

48. UNEP. Marine and Coastal Ecosystems and Human Well-being: a Synthesis Report Based on the Findings of the Millennium Ecosystem Assessment. UNEP. 2006.

49. GOSC. State Council executive meeting decides to further implement the key industry adjustment and revitalization plan (in Chinese). 2010. Available in the website. http://www.gov.cn/ldhd/2010-02/24/content_1540570.html.

50. Suo AN, Zhang MH. Sea areas reclamation and coastline change monitoring by remote sensing in coastal zone of Liaoning in China. J Coastal Res. 2015; 73:725-729.

51. Li CX, Gao X, Wu JY, Wu KN. Demand prediction and regulation zoning of urban-industrial land: evidence from Beijing-Tianjin-Hebei urban agglomeration, China. Environ Monit Assess. 2019; 191: 412, https://doi.org/10.1007/ s10661-019-7547-4.

52. Cui BS, He Q, Gu BH, Bai JH, Liu XH. China’s coastal wetlands: understanding environmental human impacts for management and conservation. Wetlands. 2016; 36: S1–S9.

53. Sousa CAM, Cunha ME, Ribeiro L. Tracking 130 years of coastal wetland reclamation in Ria Formosa, Portugal: opportunities for conservation and aquaculture. Land Use Policy. 2020; 94: 104544. https://doi.org/10.1016/j.landusepol.2020.104544.

54. Arkema KK, Guannel G, Verutes G, Wood SA, Guerry A, Ruckelshaus M, et al. Coastal habitats shield people and property from sea-level rise and storms. Nat. Clim. Change. 2013; 3: 913–918.

55. Currin C, Davis J, Baron LC, Malhotra A, Fonseca M. Shoreline Change in the New River Estuary, North Carolina: Rates and Consequences. J Coastal Res. 2015; 31(5): 1069–1077.

56. Dafforn KA, Mayer-Pinto M, Morris RL, Waltham. Application of management tools to integrate ecological principles with the design of marine infrastructure. J Environ. Manage. 2015; 158: 61–73.

57. Zhu GR, Xu XG, Wang H, Li TY, Feng Z. The ecological cost of land reclamation and its enlightenment to coast sustainable development in the northwestern Bohai Bay, China. Acta Oceanol. Sin. 2017; 36(4):97–104.

58. Zhang YZ, Chen RS, Wang Y. Tendency of land reclamation in coastal areas of Shanghai from 1998 to 2015. Land Use Policy. 2020; 91: 104370.

59. Zhang BaL. Environmental impacts of sea reclamation in Jiaozhou Gulf, Shandong province of China. Natural Hazards. 2012; 63: 1269-1272. Doi: 10.1007/s11069-012-0157-x.

60. Nicholls RJ, Wong PP, Burkett VR, Codignotto JO, Hay JE, McLean RF, et al. Coastal systems and low-lying areas. Climate change 2007: impacts, adaptation and vulnerability. In: Parry ML, Canziani OF, Palutikof JP, van der Linden PJ, Hanson CE (Eds.), Contribution of Working Group II to the Fourth Assessment Report of the Intergovernmental Panel on Climate Change. Cambridge University Press, Cambridge, UK, 2007; 315–356. 

61. Ramesh R, Chen Z, Cummins V, Day J, D’Elia C, Dennison B, et al. Land-ocean interactions in the coastal zone: past, present & future. Anthropocene. 2015; 12: 85–98.

62. Han X. 2011 National Economic and Social Development Bulletin of China’s 11 Coastal Provinces and Metropolises (in Chinese). 2012. Available in the website: http://district.ce.cn/zg/201202/27t20120227_23109765.shtml.

Additional Editor Comments (if provided):

(1) Question: The layout of the paper is very bad, especially the tables and headings.

Response: We are thankful to the Editors comments. The editor thinks the layout of the manuscript is bad and we have made improvements in tables and headings. Please refer to the revised manuscript. 

(2) Question: I suggest the results should be better discussed and justified, such as whether they are consistent with previous studies or analyzing the reasons for the empirical results. At present, there are many contents in the discussion section that are not closely related to this paper.

Response: We are thankful to the editor’s comment. The editor thinks the discussion section is not consistent with the results and we have modified the discussion based on the editor’s comments. Please refer to the revised manuscript.

Comments to the Author

(1) Question: If the authors have adequately addressed your comments raised in a previous round of review and you feel that this manuscript is now acceptable for publication, you may indicate that here to bypass the “Comments to the Author” section, enter your conflict of interest statement in the “Confidential to Editor” section, and submit your "Accept" recommendation.

Reviewer #1: (No Response)

Reviewer #2: All comments have been addressed

Response: Thank the reviewer’s high evaluation on the manuscript. The second reviewer believe that ‘All comments in the manuscript have been addressed’. We totally agree with the reviewer’s comments. During previous revision process, we took plenty of time revise the manuscript. Each comments proposed by the two reviewers has been tackled correctly. Coastal land reclamation has become a common method to mitigate land shortages in coastal area. The Bohai Sea is the most concentrated area of port reclamation worldwide. We believe that the revised manuscript has broad readers worldwide. 

(2) Question: Is the manuscript technically sound, and do the data support the conclusions? The manuscript must describe a technically sound piece of scientific research with data that supports the conclusions. Experiments must have been conducted rigorously, with appropriate controls, replication, and sample sizes. The conclusions must be drawn appropriately based on the data presented

Reviewer #1: (No Response)

Reviewer #2: Yes

Response: We are thankful to the two reviewers for the helpful comment. The Reviewer #2 think that the manuscript technically sound and the data support the conclusion. The revised manuscript used the Landsat remote sensing images and field investigation to address the changes in different type coastlines and the CLR process of these ports in the reclaimed areas of the Bohai Sea. The port land reclamation experienced periods of acceleration, peak, deceleration, and stagnation since 2002, and the planned schedule of reclamation should be constrained. The results obtained in the manuscript have important implications for further scientifically reclamation, and promoting regional sustainable development of coastal zones in dense port distribution.

 (3) Question: Has the statistical analysis been performed appropriately and rigorously?

Reviewer #1: (No Response)

Reviewer #2: Yes

Response: We are thankful to the reviewer’s evaluation on the manuscript. The revised manuscript has used the statistical method to identify the reclamation status appropriately. We believe the latest revised manuscript should meet the requirements of the journal. The readers will have no difficulties in understanding the ideas of the manuscript. We express our gratitude's to the two reviewer for their helpful comments.

 (4) Question: Have the authors made all data underlying the findings in their manuscript fully available?

Reviewer #1: (No Response)

Reviewer #2: Yes

Response: We are thankful the reviewer’s assessment toward the manuscript. We have known that the PLoS ONE Data policy requires the authors to make all data underlying the findings described in their manuscript fully available without restriction. All the images data are free download in the website. Therefore, the data presented in the manuscript has been provided as part of the manuscript. We express our gratitude's to the reviewer for the helpful comments.

 (5) Question: Is the manuscript presented in an intelligible fashion and written in standard English? PLOS ONE does not copyedit accepted manuscripts, so the language in submitted articles must be clear, correct, and unambiguous. Any typographical or grammatical errors should be corrected at revision, so please note any specific errors here.

Reviewer #1: (No Response)

Reviewer #2: Yes

Response: We are thankful to the reviewer’s assessment toward the manuscript. The manuscript is presented in an intelligible fashion and written in a standard expression. The writing expression has been revised by a Native English Speaker carefully. There is no any typographical or grammatically errors. Therefore, we believe the language in current version is clear, correct, and unambiguous. Please refer to the revised manuscript. Thank you again. 

Review Comments to the Author. 

Reviewer #1: 

(1) Question: Although the author has done a lot of revision work, there are still many problems and I cannot recommend it for publication at the present version.

1. The layout of the paper is very bad, especially the tables and headings.

Response: We are thanking the reviewer’s evaluation on the manuscript. The reviewers said that the author has done a lot of revision work. However, there are still many problems that need further revision. Port reclamation is the main types of coastal reclamation. However, we should not blame the port reclamation for several reasons. First, not all the reclaimed area is used as port construction and only 26.3% of reclaimed areas were used as port construction in the Bohai Sea.

The reviewer think that the layout of the paper is bad, especially the tables and headings. We have modified the Tables and Headings based on the reviewer’s comments. Please refer to the revised manuscript. 

(2) Question: I suggest the results should be better discussed and justified, such as whether they are consistent with previous studies or analyzing the reasons for the empirical results. At present, there are many contents in the discussion section that are not closely related to this paper.

Response: We are thankful to the reviewer’s comments. The reviewer thinks that the results should be better discussed and justified. The discussion section in the revised manuscript is consistent with previous studies. As for the coherency with the discussion, we have modified the discussion section and related information has been added in the revised manuscript. Some information that is not closely related to the paper has been deleted in the revised manuscript. 

Reviewer #2: 

(1) Question: This manuscript has been revised in detail according to the comments of the reviewers. The current version is worth publishing.

Response: We are thankful to the reviewer and the reviewer think the current version is worth publishing. The reviewer thinks that the manuscript has been revised in detail according to the comments of the reviewers. The revised manuscript takes the Bohai Sea as an example to demonstrate the coastline length change, changes of area percentage of port reclamation within the port boundary of different ports. The purpose of the revised manuscript is not to agree with the reclamation in China, but to identify the area percentage of port reclamation, i.e., used for port purpose, in the reclaimed area. The manuscript has vital meanings for decision-makers to revise the reclamation policy in future. The ports will play an increasingly important role in the world due to the frequent exchanges of materials worldwide. We express our gratitude to the warm-hearted reviewer for the constructive comments.

---

## [Decision Letter · Decision Letter 2]

21 Oct 2021

Land reclamation pattern and environmental regulation guidelines for port clusters in the Bohai Sea, China

PONE-D-21-22241R2

Dear Dr. Xie,

We’re pleased to inform you that your manuscript has been judged scientifically suitable for publication and will be formally accepted for publication once it meets all outstanding technical requirements.

Kind regards,

Jun Yang

Academic Editor

PLOS ONE

Additional Editor Comments (optional):

Accept

Reviewers' comments:

Reviewer's Responses to Questions

**Comments to the Author**

1. If the authors have adequately addressed your comments raised in a previous round of review and you feel that this manuscript is now acceptable for publication, you may indicate that here to bypass the “Comments to the Author” section, enter your conflict of interest statement in the “Confidential to Editor” section, and submit your "Accept" recommendation.

Reviewer #1: (No Response)

2. Is the manuscript technically sound, and do the data support the conclusions?

Reviewer #1: (No Response)

3. Has the statistical analysis been performed appropriately and rigorously? 

Reviewer #1: (No Response)

4. Have the authors made all data underlying the findings in their manuscript fully available?

Reviewer #1: (No Response)

5. Is the manuscript presented in an intelligible fashion and written in standard English?

Reviewer #1: (No Response)

6. Review Comments to the Author

Reviewer #1: The authors have adequately addressed the comments raised in a previous round of review and I feel that this manuscript is now acceptable for publication.

7. PLOS authors have the option to publish the peer review history of their article (what does this mean?). If published, this will include your full peer review and any attached files.

Reviewer #1: No

---

## [Editor Report · Acceptance letter]

25 Oct 2021

PONE-D-21-22241R2 

Land reclamation pattern and environmental regulation guidelines for port clusters in the Bohai Sea, China 

Dear Dr. Xie:

I'm pleased to inform you that your manuscript has been deemed suitable for publication in PLOS ONE. Congratulations! Your manuscript is now with our production department. 

Kind regards, 

on behalf of

Dr. Jun Yang 

Academic Editor

PLOS ONE